# Structural basis for substrate recognition and chemical inhibition of oncogenic MAGE ubiquitin ligases

Seung Wook Yang[1,6], Xin Huang [1,6], Wenwei Lin[2], Jaeki Min [2], Darcie J. Miller[3], Anand Mayasundari[2], Patrick Rodrigues[4], Elizabeth C. Griffith[2], Clifford T. Gee[2], Lei Li[5], Wei Li [5], Richard E. Lee [2], Zoran Rankovic[2], Taosheng Chen [2] & Patrick Ryan Potts [1✉]

Testis-restricted melanoma antigen (MAGE) proteins are frequently hijacked in cancer and play a critical role in tumorigenesis. MAGEs assemble with E3 ubiquitin ligases and function as substrate adaptors that direct the ubiquitination of novel targets, including key tumor suppressors. However, how MAGEs recognize their targets is unknown and has impeded the development of MAGE-directed therapeutics. Here, we report the structural basis for substrate recognition by MAGE ubiquitin ligases. Biochemical analysis of the degron motif recognized by MAGE-A11 and the crystal structure of MAGE-A11 bound to the PCF11 substrate uncovered a conserved substrate binding cleft (SBC) in MAGEs. Mutation of the SBC disrupted substrate recognition by MAGEs and blocked MAGE-A11 oncogenic activity. A chemical screen for inhibitors of MAGE-A11:substrate interaction identified 4-Aminoquinolines as potent inhibitors of MAGE-A11 that show selective cytotoxicity. These findings provide important insights into the large family of MAGE ubiquitin ligases and identify approaches for developing cancer-specific therapeutics.

[1] Department of Cell and Molecular Biology, St. Jude Children's Research Hospital, 262 Danny Thomas Pl, Memphis, TN 38105, USA. [2] Department of Chemical Biology and Therapeutics, St. Jude Children's Research Hospital, 262 Danny Thomas Pl, Memphis, TN 38105, USA. [3] Department of Structural Biology, St. Jude Children's Research Hospital, 262 Danny Thomas Pl, Memphis, TN 38105, USA. [4] Hartwell Center for Bioinformatics and Biotechnology, St. Jude Children's Research Hospital, 262 Danny Thomas Pl, Memphis, TN 38105, USA. [5] Division of Computational Biomedicine, Department of Biological Chemistry, School of Medicine, University of California Irvine, 5270 California Ave, Irvine, CA 92617, USA. [6] These authors contributed equally: Seung Wook Yang, Xin Huang. ✉email: Ryan.Potts@stjude.org

Melanoma antigens (MAGEs) are a large (>40 in humans) and highly conserved family of proteins present in all eukaryotes[1,2]. MAGEs can be classified into two categories based on their tissue expression pattern[2,3]. Two-thirds of MAGEs, including MAGE-A, -B, and -C subfamilies are considered type I MAGEs due to their restricted expression in the testis and other reproductive tissues[3–6]. Conversely, MAGE-D, -E, -F, -G, -H, -L and Necdin subfamilies are type II MAGEs that have broad expression in many tissues[1,2]. Both type I and II MAGEs share a conserved domain known as the MAGE homology domain (MHD) involved in protein–protein interactions. MHDs consist of tandem winged-helix motifs (WH-A and WH-B) that feature a helix-turn-helix packed against a three-stranded antiparallel beta-sheet[7–9]. The overall structures of MHDs are similar between MAGEs, but the relative orientation of WH-A to WH-B can differ[8,9].

Although type I MAGEs are typically restricted to expression in the testis, they are often aberrantly expressed in many cancer types and are often referred to as cancer-testis antigens[1,3]. The aberrant expression of MAGEs in tumors is not simply an inert consequence of genomic dysregulation, but instead MAGEs play active roles in driving tumorigenesis[1,5,10–13]. For example, MAGE-A3, -A6, and -A11 are aberrantly expressed in many cancer types, including melanoma, breast, colon, and lung cancers, where they are required for cancer cell viability and promote multiple hallmarks of cancer, such as anchorage-independent growth and xenograft tumor growth[1,5,10,14]. Recent studies have highlighted an important role of MAGEs in controlling core oncogenic and tumor suppressor pathways in cells, including metabolic rewiring by MAGE-A3/6 through modulation of AMPK and autophagy and promotion of alternative polyadenylation by MAGE-A11[5,10]. In addition, MAGE-F1 is highly amplified in multiple cancer types leading to reduced DNA repair capacity and increased mutational burden in tumors through downregulation of the cytosolic iron-sulfur assembly pathway[15]. Thus, given the prominent, cancer-selective expression of MAGEs and their oncogenic functions, MAGEs have emerged as prime therapeutic targets. However, this effort has been impeded by our limited understanding of how to target these enigmatic proteins.

Growing evidence suggests that MAGEs function as substrate adapters for E3 ubiquitin ligases (reviewed in Lee and Potts[2]). MAGEs bind to specific single-subunit E3 ubiquitin ligases, both RING and HECT type ligases, including MAGE-A1-TRIM31, MAGE-A3-TRIM28, MAGE-A11-HUWE1, MAGE-B18-LNX1, MAGE-D1-PRAJA1, MAGE-F1-NSE1, MAGE-L2-TRIM27, and MAGE-G1-NSE1[8,10,16–18]. Importantly, MAGEs alter the function of these ligases by recruiting novel substrates to the ligases for ubiquitination, including MAGE-A3 mediating the ubiquitination of AMPK by TRIM28, MAGE-A11 mediating ubiquitination of PCF11 by HUWE1 and MAGE-F1 mediating ubiquitination of MMS19 by NSE1[5,10,15]. In each of these cases, the MAGE functions as protein glue that mediates recruitment of the substrate to the ligase complex for ubiquitination. Thus, MAGEs reprogram single-subunit ubiquitin ligases. Notably, the specific MAGE bound to a ligase can dictate distinct substrates and functions. For example, both MAGE-F1 and MAGE-G1 bind the NSE1 ligase. However, these MAGEs have distinct substrates with opposing cellular functions in modulating DNA repair pathways[8,15,19]. In many ways MAGEs have similar properties to Cullin-RING Ligase (CRL) adapter protein families, such as F-box, BTB, DCAF, and SOCS box proteins, which allow modularity, functional diversification, and adaptability of CRLs[20]. However, unlike CRL adapters, the molecular and structural basis of how MAGEs recognize substrates has been unknown.

Here, we determine how the MAGE-A11 oncogene recognizes its substrate, PCF11. Biochemical analysis uncover a consensus degron motif sequence recognized by MAGE-A11 in PCF11 and other putative substrates. The crystal structure of MAGE-A11 bound to its substrate PCF11 is determined revealing a conserved substrate binding cleft (SBC) in the MHD. Importantly, mutational analysis of this region across MAGEs reveals its importance for MAGE:substrate recognition with sequence diversity around the SBC driving MAGE substrate selectivity. Disruption of the MAGE-A11 SBC blocks MAGE-A11 oncogenic activity suggesting an avenue for discovery of MAGE-directed cancer therapeutics. An in vitro biochemical assay is first developed to monitor and map the protein–protein interaction between MAGE-A11 and PCF11, which is then used in a high-throughput chemical screen to discover sub-micromolar small molecule inhibitors of MAGE-A11 that lead to MAGE-dependent cytotoxicity. These findings advance our understanding of MAGE ubiquitin ligases and provide proof of concept for chemical inhibition of MAGEs in the pursuit of cancer therapeutics.

## Results

**Characterization of MAGE-A11 interaction with PCF11.** Recently, we reported that MAGE-A11 drives tumorigenesis through ubiquitination of the 3′-mRNA processing factor PCF11 resulting in alternative polyadenylation (APA) of transcripts and 3′-UTR shortening (3′-US)[10]. To understand how MAGE-A11 recognizes its substrate, we mapped the degron motif in PCF11 required for MAGE-A11 binding. We found that deletion of PCF11 amino acids 653-702 completely blocked MAGE-A11 binding by co-immunoprecipitation (co-IP) (Fig. 1a). To narrow down the minimal degron motif within these 50 amino acids, we developed a time-resolved fluorescence resonance energy transfer (TR-FRET) assay to monitor binding of a series of PCF11 peptides to GST-MAGE-A11 in vitro (Fig. 1b and Supplementary Fig. 1a). Notably, of the eight PCF11 peptides tested, three overlapping peptides centered on PCF11 amino acids 682–696 showed robust binding with $IC_{50}$ of 59–77 nM (Fig. 1c). These findings were confirmed by an orthogonal assay showing changes in the thermal stability of MAGE-A11 by the PCF11 degron peptide (Fig. 1d and Supplementary Fig. 1b).

To identify the specific residues in the PCF11 degron responsible for interaction with MAGE-A11, we performed alanine scanning mutagenesis of the PCF11 degron peptide and monitored binding to MAGE-A11 by TR-FRET (Supplementary Fig. 1c). Mutation of F682, I689, R690, L692, or F693 to alanine dramatically decreased binding 10–200 fold, with other positions having minimal effects (Fig. 1e and Supplementary Fig. 1d). These findings are consistent with our previous observations that mutation of PCF11 I689A blocked PCF11 interaction with MAGE-A11 in cells and prevented MAGE-A11-induced xenografts tumor growth[10].

Given the success of our binding assay to map specific amino acid sequences competent for MAGE-A11 binding, we expanded our study to perform TR-FRET affinity measurements on a peptide array consisting of 248 peptides in which each position within the model 13 amino acid PCF11 peptide degron is mutated to all possible amino acids. Consistent with our alanine scanning mutagenesis, the same five amino acid positions were again identified as being the most important (Fig. 1f). Position +1 (PCF11 F682) preferred an aromatic residue, position +8 (PCF11 I689) required a hydrophobic residue, position +9 (PCF11 R690) only tolerated a basic residue, position +11 (PCF11 L692) preferred a large hydrophobic residue and position +12 (PCF11 F693) preferred aromatic residues. Overall, the degron sequence conforms to [FLWY]-$X_6$-[IV]-[KR]-X-[ILMV]-[FLWY], where residues denoted as X being tolerable to several amino acids. Notably, the PCF11 sequence is not fully optimized with several

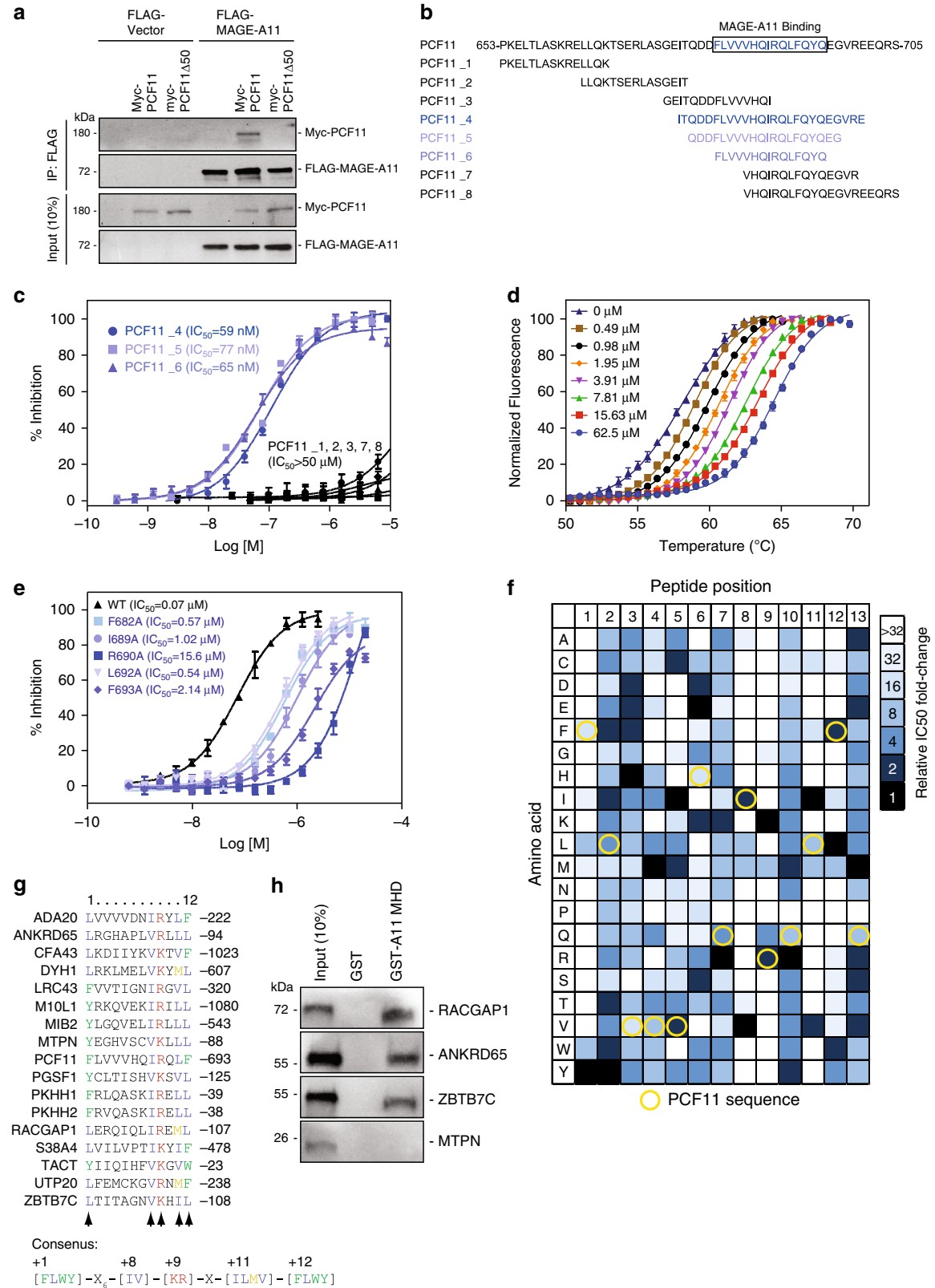

substitutions yielding higher affinity (Fig. 1f). These insights allowed us to build a consensus MAGE-A11 binding motif weighted matrix (Supplementary Fig. 1e). Search of the proteome using this matrix revealed 17 possible MAGE-A11 targets with the consensus binding motif (Fig. 1g). To validate our prediction, we examined the ability of MAGE-A11 to bind four of these

newly predicted targets in vitro and found RACGAP1, ANKRD65, and ZBTB7C to be binding partners (Fig. 1h). Notably, we did not detect interaction with MTPN (Fig. 1h), however the MAGE-A11 binding motif is not surface exposed in this protein (PDB: 3AAA). These results provide detailed understanding of the sequences recognized by MAGE-A11.

**Fig. 1 Characterization of MAGE-A11 interaction with substrates. a** The MAGE-A11 recognition sequence in PCF11 is within amino acids 653-702. HEK293FT cells stably expressing FLAG-vector or FLAG-MAGE-A11 were transfected with PCF11 wild-type or PCF11 653-702 deletion construct for 48 h followed by IP with anti-FLAG, SDS-PAGE, and immunoblotting for anti-Myc. Source data are provided as a Source Data file. **b, c** Mapping of MAGE-A11-interacting motif in PCF11 by TR-FRET. Sequences of PCF11 degron motif and tested eight PCF11 peptides are shown (**b**). The binding activity of each peptide to GST-MAGE-A11 by TR-FRET is shown (**c**) ($n = 2$ biologically independent experiments with triplicates per peptide). Data are mean ± SD. **d** Validation of PCF11 peptide binding to MAGE-A11 by thermal stability assay with $n = 3$ biologically independent experiments per concentration. Thermal unfolding of MAGE-A11 is monitored with increasing concentrations of PCF11_6 peptide. Data are mean ± SD. **e** Alanine scanning mutagenesis of PCF11_6 peptide binding to GST-MAGE-A11 by TR-FRET ($n = 2$ biologically independent experiments with triplicates per peptide). Data are mean ± SD. **f** Relative TR-FRET affinity measurements of PCF11 degron peptide variants library (248 peptides) is shown. Coloring indicates relative binding affinity normalized at each position. Yellow circles indicates sequence of natural PCF11 degron peptide. **g** Comparison of 16 predicted MAGE-A11-binding motifs. **h** Recombinant GST-MAGE-A11 binds in vitro translated Myc-RACGAP1, ANKRD65, and ZBTB7C. Source data are provided as a Source Data file.

## Crystal structure of MAGE-A11 bound to PCF11. 
MAGE genes are defined by a common conserved MAGE homology domain (MHD), which is involved in protein–protein interactions[2,8,9]. Similar to other MAGE:substrate interactions[5,8], PCF11 binding to MAGE-A11 is mediated through the MAGE-A11 MHD (197–429; Fig. 2a). To understand how substrates interact with MAGEs, we determined the crystal structure of MAGE-A11 bound to the PCF11 degron. We initially attempted to determine the co-crystal structure of MAGE-A11 MHD bound to free PCF11 degron supplied as a peptide post purification of MAGE-A11 MHD from bacteria. However, this approach did not yield homogenous protein:peptide complex, likely due to the MAGE-A11 MHD apo-protein being less stable during expression and purification from bacteria. Thus, we expressed and purified the PCF11 degron sequence fused to the MAGE-A11 MHD (S218 to C-terminus) from bacteria and solved the crystal structure to 2.2 Å (Table 1). MAGE-A11 MHD formed two winged-helix motifs (WH-A and WH-B) conjugated by a linker loop, where each WH motif contains three alpha-helices and two beta-sheets. The overall MAGE-A11 structure is similar to the other two reported MAGE-A family protein structures (MAGE-A3-MHD PDB: 4V0P and MAGE-A4-MHD PDB: 2WA0). However, the extreme C-terminal residues of MAGE-A11 formed three alpha-helices, whereas these residues are unstructured in MAGE-A3-MHD and MAGE-A4-MHD structures (Fig. 2b)[9]. Additionally, no evidence of MAGE-A11 dimer formation was observed in the crystal structure or in cell lysates by co-IP (Supplementary Fig. 2a). Of particular interest, the PCF11 degron forms a short alpha-helix (Fig. 2b, c and Supplementary Fig. 2b, c) that binds into a shallow cleft (Fig. 2c) formed at the interface of the WH-A and WH-B motifs. We refer to this region on MAGE-A11 as the substrate binding cleft (SBC).

PCF11 binding to the MAGE-A11 SBC is largely mediated by numerous hydrophobic interactions that extend along its helical core and capping residues (Fig. 2d and Supplementary Fig. 2d). I689 and L692 of PCF11 form a hydrophobic core with L273, L274, I317, M341, V337, I340, and L338 of MAGE-A11. Notably, the N-terminal capping residue F682 of PCF11 interacts with F275, F350, L351, H231, and V230, whereas F693 at the C-terminus interacts with I317, I403, and A406 in MAGE-A11. In addition, a key polar interaction network enhances binding; R690 in PCF11 forms a water-mediated salt bridge with E266 in MAGE-A11 and is further stabilized by forming a hydrogen bond between the NE2 moiety and the backbone carbonyl of A406. Taken together, these structural observations are highly consistent with our TR-FRET mutagenesis data. A detailed description of polar and hydrophobic PCF11 interactions with MAGE-A11 is provided in Supplementary Table 1.

Next, we sought to validate the MAGE-A11:PCF11 structure through mutation of the MAGE-A11 SBC and measuring PCF11 binding in vitro and in cells. Mutation of MAGE-A11 SBC residues F275 or Y414 to alanine significantly decreased MAGE-A11 interaction with PCF11 in vitro (Fig. 2e). These findings were corroborated by co-IP studies in cells (Fig. 2f). Furthermore, disruption of the hydrophobic core of the MAGE-A11 SBC by I317R or M341R ablated PCF11 interaction in vitro (Fig. 2e) and in cells (Fig. 2f). In comparison, mutation of MAGE-A11 H231 that is not involved in PCF11 recognition had no effect on PCF11 binding in vitro (Fig. 2e). These results are consistent with the MAGE-A11:PCF11 crystal structure suggesting that the MAGE-A11 SBC serves as a critical region for substrate recognition. Importantly, mutation of the MAGE-A11 SBC did not disrupt binding to the MAGE-A11 associated E3 ligase HUWE1 (Fig. 2f). Together, these results suggest that the MAGE-A11 SBC mediates substrate interaction and E3 ligase binding occurs independent of the MAGE-A11 SBC.

Given the importance of the C-terminal end of the PCF11 motif (position #8, #9, #11, and #12) for MAGE-A11 binding and that MAGE-A11:PCF11 interaction affinity can be increased by amino acid substitutions (Fig. 1f), we synthesized several peptides with non-natural phenylalanine derivatives at PCF11 F693 (position #12). This included F, Cl, $CF_3$, CN, or $CH_3$ substitutions at the 2, 3, or 4 position of the phenylalanine benzene ring (Fig. 2g and Supplementary Fig. 2e). One variant, 3-methyl-phenylalanine, significantly increased PCF11 binding to MAGE-A11 (Fig. 2g; $IC_{50} = 1.7$ nM), likely due to increased hydrophobic and Van der Waals contacts. However, substitutions at the 4th position on the benzene ring, including 4-cyano- phenylalanine (Fig. 2g; $IC_{50} = 663$ nM), were not tolerated well, likely due to steric hindrance. Overall, these findings provide structural basis of MAGE-A11 binding to PCF11 and suggest mechanisms to improve MAGE-A11:PCF11 substrate binding affinity.

## Mutation of MAGE-A11 SBC disrupts its oncogenic activity. 
To determine whether mutation of the MAGE-A11 SBC disrupts its ability to ubiquitinate PCF11 and control tumor cell growth, we stably reconstituted MAGE-A11 knockout DAOY cells with MAGE-A11 SBC mutants F275A or M341R and assessed PCF11 ubiquitination and subsequent effects on cell growth and tumor formation. Consistent with disruption of PCF11 binding by MAGE-A11 SBC mutation (Fig. 2f), MAGE-A11 F275A and M341R failed to promote PCF11 ubiquitination in cells (Fig. 3a). Moreover, unlike wild-type MAGE-A11, MAGE-A11 F275A, and M341R failed to promote proliferation of MAGE-A11 knockout DAOY cells (Fig. 3b). Compared to wild-type MAGE-A11, re-expression of MAGE-A11 F275A or M341R reduced clonogenic growth (Fig. 3c). To test the importance of the MAGE-A11 SBC for MAGE-A11 oncogenic activity in vivo, we performed xenograft tumor growth assays using the MAGE-A11 knockout DAOY cells reconstituted with wild-type, F275A or M341R MAGE-A11. Consistent with MAGE-A11 oncogenic activity, wild-type MAGE-A11 promoted increased tumor growth (Fig. 3d, e). However, DAOY cells reconstituted with MAGE-A11

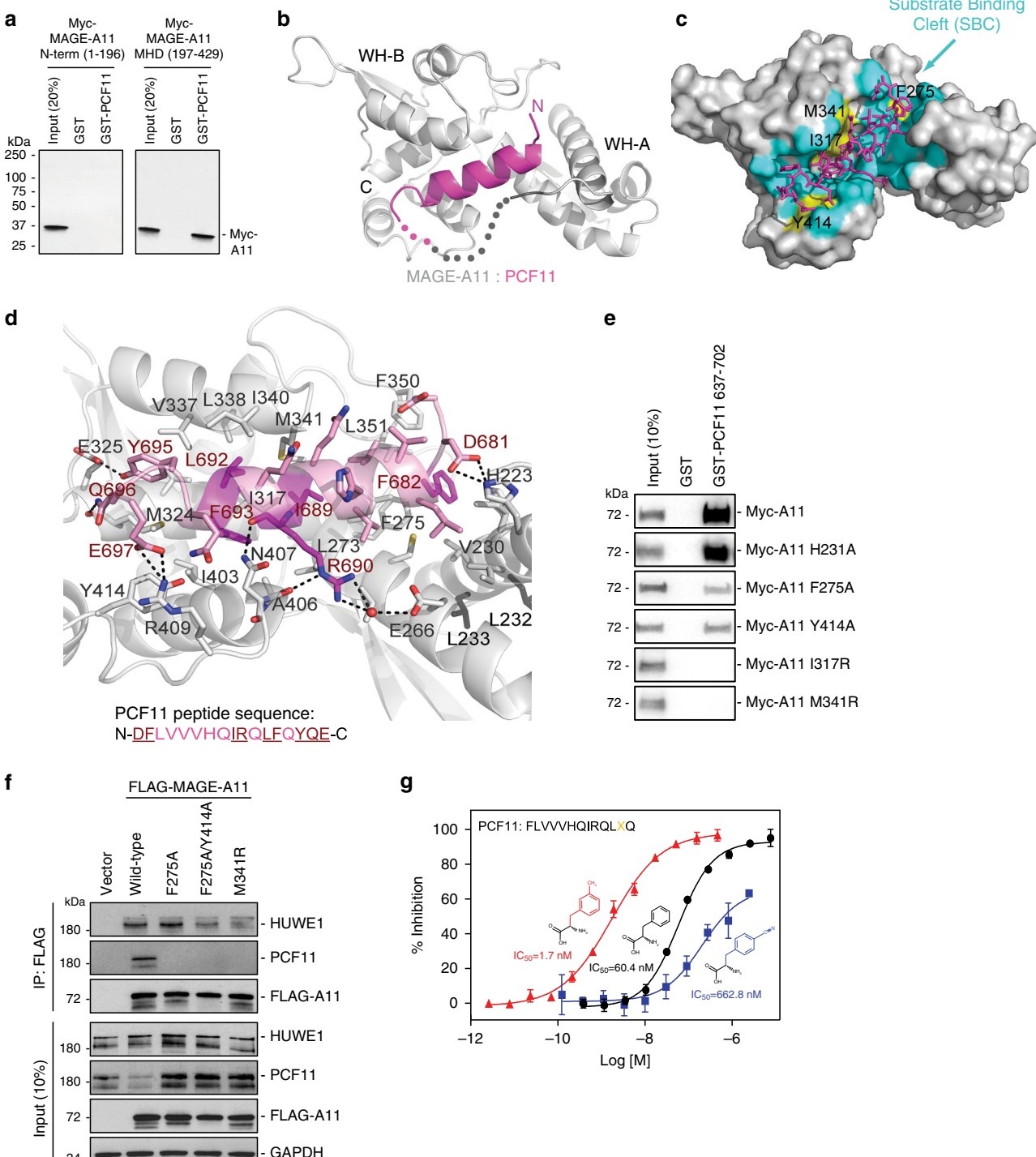

**Fig. 2 Crystal structure of MAGE-A11:PCF11 identifies MAGE substrate binding cleft. a** MAGE-A11 directly interacts with PCF11 through its MHD. In vitro translated Myc-MAGE-A11 MHD, but not N-terminal fragment, binds to recombinant GST-PCF11. Source data are provided as a Source Data file. **b** Overall structure of human MAGE-A11 MHD with human PCF11 degron peptide. The alpha-helical PCF11 degron peptide binds into SBC at interface of WH-A and WH-B motifs of MAGE-A11 MHD. PCF11 peptide is shown in pink and MAGE-A11 in gray. **c** MAGE-A11 atoms within 5 Å of PCF11 are colored cyan to indicate the SBC region. Residues in the SBC that are mutated in this study are labeled and colored yellow. **d** Substrate interaction is largely achieved by hydrophobic interactions along the alpha-helical interface of PCF11 with MAGE-A11 SBC. **e** Myc-tagged MAGE-A11 wild-type or mutants were in vitro translated and followed by in vitro binding assay with recombinant GST-PCF11 637-702 amino acids, SDS-PAGE and immunoblotting for anti-Myc. Source data are provided as a Source Data file. **f** MAGE-A11 SBC is crucial for PCF11 substrate binding, but not HUWE1 E3 ligase binding, in cells. HEK293FT cells stably expressing FLAG-vector, FLAG-MAGE-A11 wild-type or mutants were subjected to pull-down with anti-FLAG followed by SDS-PAGE and immunoblotting for anti-HUWE1 and anti-PCF11. Source data are provided as a Source Data file. **g** TR-FRET binding assays of GST-MAGE-A11 with PCF11 peptide (FLVVVHQIRQLF*Q) containing phenylalanine (black circles), 3-methyl-phenylalanine (red triangles) or 4-cyano-phenylalanine (blue squares) at the indicated position ($n = 2$ biologically independent experiments with triplicates per peptide). Data are mean ± SD.

**Table 1 Data collection and refinement statistics.**

|  | MAGE-A11:PCF11 |
|---|---|
| Data collection |  |
| Space group | P 1 |
| Cell dimensions |  |
| a, b, c (Å) | 55.50 78.12 72.91 |
| α, β, γ (°) | 77.44 77.34 89.59 |
| Resolution (Å) | 38.79-2.19 (2.24-2.19)[a] |
| $R_{sym}$ | 0.120 (0.372) |
| $I/\sigma$ (I) | 19.9 (2.4) |
| $CC_{1/2}$ | 0.952 (0.757) |
| Completeness (%) | 96.6 (94.7) |
| Redundancy | 2.5 (2.3) |
| Refinement |  |
| Resolution (Å) | 38.79-2.19 |
| No. reflections | 53723 |
| $R_{work}/R_{free}$ | 0.205/0.246 |
| No. atoms |  |
| Protein | 7085 |
| Water | 208 |
| B factors (Å²) |  |
| Protein | 40.12 |
| Water | 45.27 |
| R.M.S. deviations |  |
| Bond lengths (Å) | 0.008 |
| Bond angles (°) | 1.48 |
| Ramachandran statistics (%) |  |
| Favored | 96.38 |
| Allowed | 3.62 |
| Outliers | 0 |

[a]Values in parentheses are for the highest resolution shell.

SBC mutants failed to promote tumor growth and were indistinguishable from vector control cells (Fig. 3d, e). To determine if MAGE-A11 M341R fails to promote 3′-UTR shortening, tumors were isolated and transcriptome analysis was performed. Consistent with MAGE-A11 M341R failing to bind to and ubiquitinate PCF11, MAGE-A11 M341R tumors had significantly lengthened 3′-UTRs compared to wild-type MAGE-A11 (Fig. 3f, g). Together, these results suggest that the MAGE-A11 SBC is required for its ability to regulate PCF11, promote alternative polyadenylation and drive tumor growth.

**MAGE SBC mediates substrate recognition and specificity.** In humans there are at least 40 distinct MAGE genes that each contain at least one broadly conserved MHD, with some MHDs being >90% identical. However, not all MAGEs are redundant and their MHDs have been shown to recognize distinct substrates, including MAGE-A11:PCF11, MAGE-A3:AMPK, and MAGE-F1:MMS19[5,10,15]. Thus, specific sequence elements in the MHD likely dictate substrate specificity. Mapping sequence conservation of MAGEs to the MAGE-A11:PCF11 structure revealed that the majority of residues that are highly conserved across MAGEs are on the MHD backside away from PCF11 binding site (Fig. 4a). However, the most variable residues are concentrated around the MAGE SBC, suggesting that these are likely key positive and negative selection filters that dictate substrate specificity across the MAGE protein family. This is exemplified in comparing the SBC of MAGE-A11 with -A3 (Fig. 4b). Although the MHD structures are highly similar (RMSD 1.07; MAGE-A3 (108-293) to MAGE-A11 (221-406)), detailed examination of the SBCs revealed important differences. For example, PCF11 R690 forms a salt bridge with E266 in MAGE-A11 SBC (Fig. 4b). However, in the MAGE-A3 SBC this residue has undergone a charge swap to lysine (Fig. 4b and Supplementary Fig. 3). This along with other sequence variation in the SBC likely dictate substrate specificity for MAGE ligases. Consistent with these observations, the ability to bind PCF11 is highly specific to MAGE-A11 with none of the other ten MAGE-A proteins having detectable binding (Fig. 4c), despite overall sequence conservation between these MHDs being quite high (>80% similarity).

To determine whether other MAGEs utilize a similar SBC formed at the interface of WH-A and WH-B to mediate substrate binding, we mutated this region in MAGE-A3 and -F1 and determined their ability to bind their respective substrates, AMPK and MMS19. We found that mutation of key interaction residues in MAGE-A3 SBC (F162A, I204R, L228R, or Y301A) and MAGE-F1 SBC (F129A, L196R, or W269A) that correspond to those mutated in the MAGE-A11 SBC (Fig. 4d, f and Supplementary Fig 3) disrupted their ability to bind AMPK and MMS19 substrates, respectively (Fig. 4e, g). Additionally and consistent with MAGE-A11, mutation of MAGE-A3 SBC or MAGE-F1 SBC did not impair interaction with their cognate E3 ligase binding partners, TRIM28 and NSE1, respectively (Fig. 4e, g). Consistent with this, di-leucine residues previously shown to contribute to ligase binding do not occupy the SBC (Fig. 2d)[8]. Together, these results suggest that the mechanism for substrate recognition is conserved across MAGE proteins and is mediated the MHD SBC.

**Discovery of small molecule inhibitors targeting MAGE-A11.** MAGE-A11 expression is typically restricted to reproductive tissues, but is aberrantly expressed in many tumor types where its expression correlates with poor patient prognosis[10,14,21–23]. Moreover, MAGE-A11 functions as an oncogene and its genetic ablation inhibits tumorigenesis[10,14]. Thus, the cancer-specific MAGE-A11 E3 ubiquitin ligase is an ideal therapeutic target. Our findings suggest that disruption of MAGE-A11 interaction with PCF11 may be a viable strategy to design cancer-specific therapeutics. Although disruption of protein–protein interactions are difficult, several examples have emerged, including a small molecule agonist that inhibits T-cell proliferation through the inhibition of IL-2 binding to its receptor IL2Rα[24], small molecule inhibitors of the anti-apoptotic proteins Bcl-2, Bcl-$X_L$, and Bcl-w[25] and small molecule antagonists block HDM2 binding to p53[26].

To identify inhibitors of MAGE-A11, we performed a TR-FRET-based high-throughput screen of 31,407 compounds (summarized in Supplementary Fig. 4a) to identify small molecules that block MAGE-A11 binding to PCF11 in vitro. From this primary screen, 205 compounds (0.65% hit rate, $Z' = 0.82$) that had >30% inhibition (Supplementary Fig. 4b, c) were cherry-picked for follow-up dose response screening. A $Z'$-factor value >0.5 indicated that the screen is robust[27]. After hit clustering and structure-activity relationship (SAR) analysis from our primary and secondary screening results, we assayed an additional 500 analogs of nine scaffold groups obtained by substructure search of an in-house library collection (Fig. 5a and Supplementary Data 1). Of these scaffolds, the 145 quinoline analogs (Scaffold-4 in Supplementary Data 1) showed the most significant effects on MAGE-A11:PCF11 binding with reasonable SAR and range of activity (Fig. 5a). To expand SAR of the quinoline series, we extended diversity around the scaffold and conducted a single point TR-FRET screen of 797 additional quinoline analogs followed by dose response experiments for 59 compounds (>30% inhibition at 15 μM) (Supplementary Fig. 4d and Supplementary Data 2 and 3). From these primary and secondary screens, one of the 4-Aminoquinoline analogs, **SJ521054**, emerged as a promising hit with robust activity in

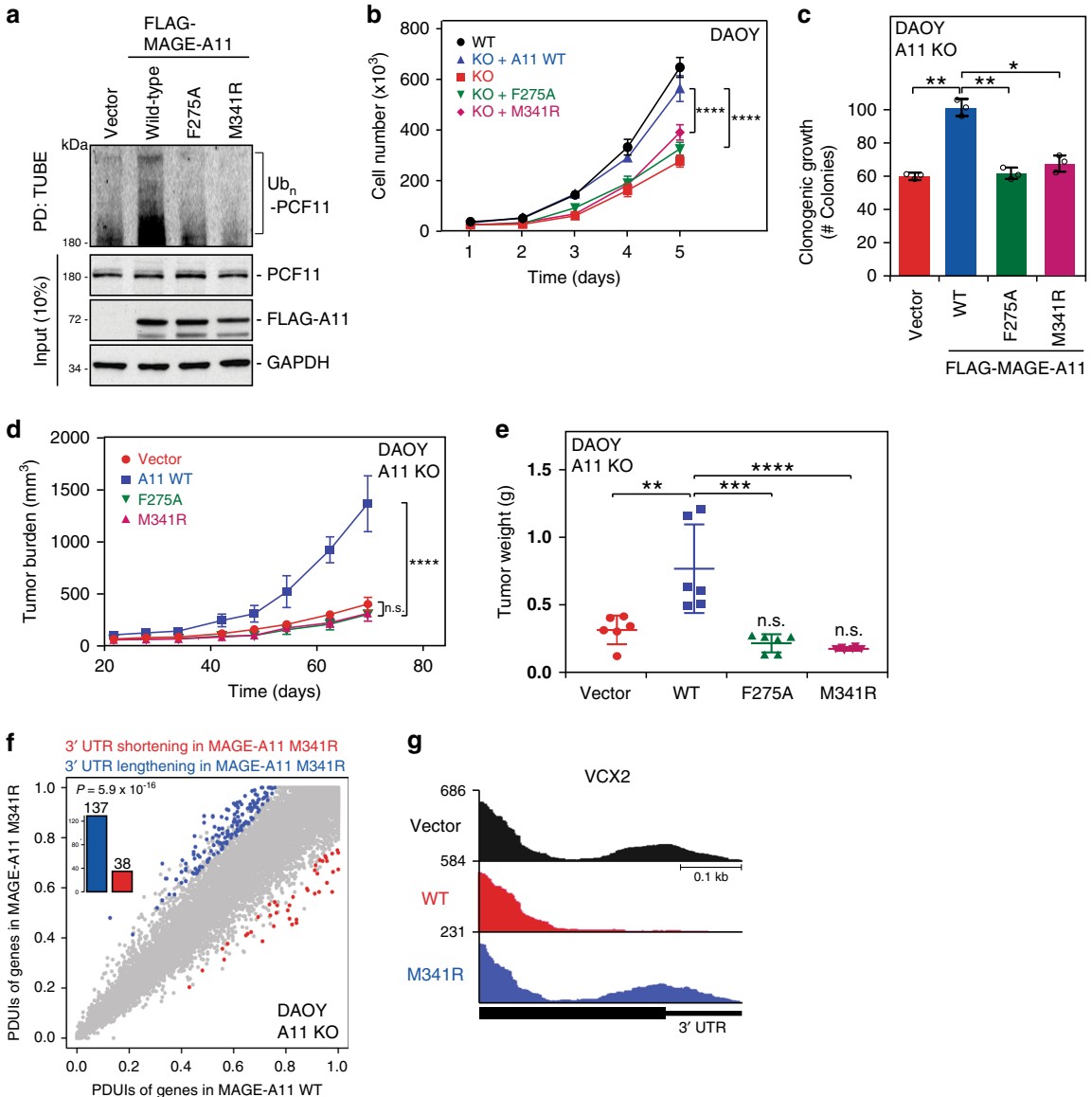

**Fig. 3 Mutation of MAGE-A11 SBC disrupts its oncogenic activity. a** MAGE-A11-induced ubiquitination of PCF11 depends on its substrate recognition by MAGE-A11 SBC. Ubiquitinated proteins from FLAG-vector, FLAG-MAGE-A11 wild-type, or SBC mutants stably expressing HEK293FT cells were treated with 10 μM MG132 for 4 h and isolated with tandem ubiquitin binding entity-agarose followed by SDS-PAGE and immunoblotting for endogenous PCF11. Source data are provided as a Source Data file. **b** MAGE-A11 SBC mutants fail to promote proliferation of MAGE-A11 knockout DAOY cells. Wild-type, MAGE-A11-knockout DAOY cells, or those reconstituted with MAGE-A11 were counted for cell proliferation at the indicated time points ($n = 3$ biologically independent experiments per group). Data are mean ± SEM. p-value by two-way ANOVA followed by Tukey's multiple comparisons test is indicated (****$P < 0.0001$). **c–e** MAGE-A11 knockout DAOY cells reconstituted with MAGE-A11 SBC mutants reduced clonogenic growth and xenograft tumor growth of MAGE-A11 knockout DAOY cells. Stable expression of wild-type MAGE-A11 or SBC mutants in MAGE-A11-knockout DAOY cells were assayed for clonogenic growth (**c**) ($n = 3$ biologically independent experiments per group, two-tailed unpaired Student's t test, *$P < 0.05$; **$P < 0.01$), Data are mean ± SEM and xenograft tumor growth in mice ($n = 6$ per group). Tumor burden (**d**) and tumor weight (**e**) are shown. Data are mean ± SD. p-value by two-way ANOVA followed by Tukey's multiple comparisons test is indicated in **d** (****$P < 0.0001$), ordinary one-way ANOVA followed by Sidak's multiple comparisons test is indicated in **e** (**$P < 0.01$, ***$P < 0.001$, ****$P < 0.0001$). Red circle, Vector; Blue square, MAGE-A11 WT; Green triangle, MAGE-A11 F275A; and Pink triangle, MAGE-A11 M341R. **f** MAGE-A11 M341R mutant fails to promote mRNA 3'UTR shortening. Transcriptome analysis of tumors ($n = 3$ per group) from MAGE-A11 knockout DAOY cells stably expressing FLAG-MAGE-A11 wild-type or M341R mutant was determined and scatterplot of Percentage of Distal polyA site Usage Index (PDUI) is shown. False discovery rate ≤ 0.05 and ΔPDUIs ≥ 0.2 are colored. The shift towards distal polyadenylation site is significant ($p = 5.9 \times 10^{-16}$, binomial test). **g** Representative RNA-seq density plots are shown.

TR-FRET for disrupting MAGE-A11:PCF11 interaction ($IC_{50}$ 0.40 μM; Fig. 5b, c).

To validate the activity of **SJ521054** in orthogonal assays, we examined (1) its ability to prevent MAGE-A11-induced degradation of PCF11 and (2) the cytotoxicity towards MAGE-A11-positive cells compared to isogenic MAGE-A11 negative cells.

Indeed, **SJ521054** stabilized PCF11 protein levels in MAGE-A11 expressing HEK293FT cells but had no effect on vector control cells (Fig. 5d). Strikingly, regardless of their proliferation rate, **SJ521054** decreased the viability of MAGE-A11 expressing DAOY and HEK293FT cells to a much greater extent than vector control MAGE-A11-negative cells (Fig. 5e and Supplementary Fig. 4f, g).

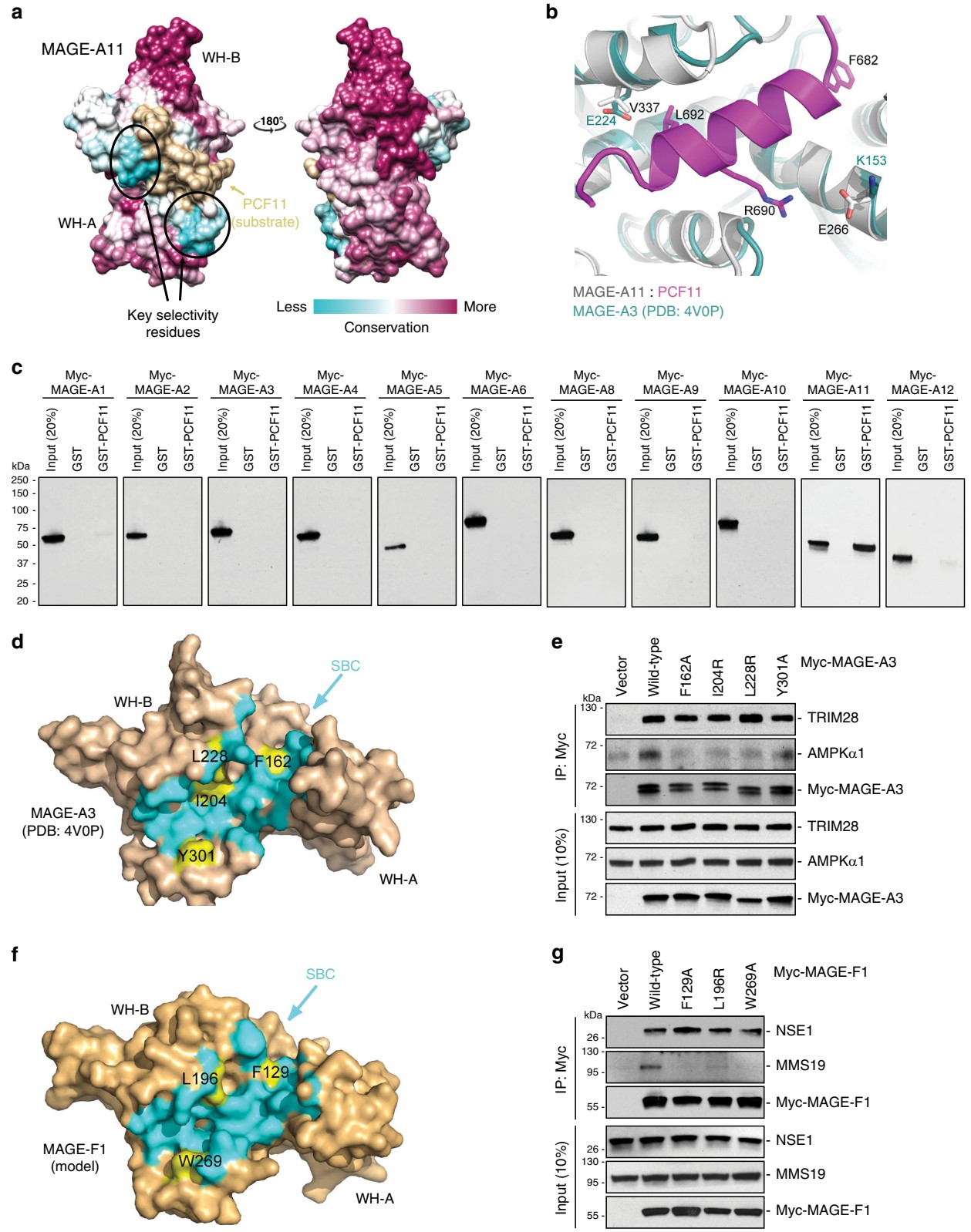

To extend upon the discovery of **SJ521054**, nine additional derivatives were designed, synthesized, and evaluated by the TR-FRET assay for their ability to disrupt MAGE-A11:PCF11 interaction and their physicochemical properties were determined along with selected previous hits for comparison (Supplementary Data 4). From these data, we identified **SJ1008066** as a lead compound that is cell permeable and has improved activity in the

TR-FRET assay ($IC_{50}$ 0.13 μM) and solubility (Fig. 5f, Supplementary Fig. 5a, and Supplementary Data 4). To evaluate direct binding of **SJ521054** and **SJ1008066** to MAGE-A11 MHD, we performed Nano-differential scanning fluorimetry (NanoDSF) assay (Supplementary Fig. 5b, c). Both compounds stabilized MAGE-A11 in a six-point does response confirming on-target activity (Fig. 5g and Supplementary Fig. 5b, c). Furthermore,

**Fig. 4 The SBC region mediates substrate recognition and specificity across the MAGE family. a** Sequence conservation of MAGE-A genes mapped onto MAGE-A11:PCF11 structure. **b** Comparison of MAGE-A3 and -A11 SBC reveals key sequence differences in critical substrate binding residues. **c** In vitro GST-pulldown assays reveals that GST-PCF11 binds in vitro translated Myc-MAGE-A11, but no other Myc-MAGE-A protein. Source data are provided as a Source Data file. **d–g** Mutation of MAGE-A3 or -F1 SBC disrupts binding to their respective substrates AMPK and MMS19, not the E3 ligases TRIM28 and NSE1. Surface representations of MAGE-A3 (protomer model formed by PDB ID 4V0P for residues 1-296 and MAGE-A11 based homology model for C-terminal residues 297–311) (**d**) and MAGE-F1 (SWISS-MODEL based on PDB:4V0P) (**f**) show positioning of SBC residues mutated. Consistent with Fig. 2c, MAGE atoms within 5 Å of modeled PCF11 (based on superimposition of MAGE-A11:PCF11) are colored cyan. Residues in the SBC that are mutated are labeled and colored yellow. Co-IP of MAGEs with substrates and ligases were determined in HeLa cells (**e**) or HEK293FT cells (**g**) upon transfection of the indicated constructs for 48 h, IP with anti-Myc, SDS-PAGE, and immunoblotting for indicated proteins. Source data are provided as a Source Data file.

**SJ521054** and **SJ1008066** inhibited MAGE-A11 binding to PCF11 in cell lysates as determined by co-IP, with much less potent compounds having minimal effects (Fig. 5h and Supplementary Fig. 4e). Interestingly, both compounds did not disrupt interaction of MAGE-A3 with AMPKα1 or MAGE-F1 with MMS19 (Fig. 5i). Taken together, these results identify tool compounds that provide proof-of-principle that specific disruption of MAGE-A11:substrate interaction may be a means to discover cancer-specific therapeutics.

## Discussion

MAGE genes were discovered almost three decades ago as reproductive tissue restricted genes that when aberrantly expressed in cancer elicit a cytotoxic immune response due to their antigenicity[28]. This discovery set off a flurry of research on the potential of MAGEs to prime the immune systems of cancer patients to target tumor cells[29]. However, the underlying function and biochemical activity of MAGE proteins were largely understudied. More recently, greater understanding into the normal physiological and disease functions of MAGEs has been elucidated[1–3]. Importantly, the MAGE family of proteins were biochemically defined as scaffolding proteins that diversify the function of E3 ubiquitin ligases through the recruitment of novel substrates for ubiquitination[2,8]. However, many questions remained unanswered that have hindered our understanding of how MAGEs work and therapeutically targeting the many MAGE cancer-testis antigens. Chiefly, the molecular and structural basis of how MAGEs identify substrates was unknown. The specific degron sequences recognized by any given MAGE was unclear and the sequence and structural features on MAGEs that mediate recognition were unknown. These questions have important implications into MAGE biology, including understanding how relatively similar MAGE proteins can have distinct targets that impinge on a variety of cellular pathways. Answering these questions will facilitate the design of MAGE chemical inhibitors. Here, we provide the first evidence of how MAGEs recognize substrates and identify chemical inhibitors of the cancer selective, oncogenic MAGE-A11.

We have determined the crystal structure of MAGE-A11 bound to its substrate PCF11. Strikingly, a small peptide of PCF11 forms short alpha-helix that binds into a hydrophobic cleft formed at the interface of WH-A and WH-B motifs within the MAGE-A11 MHD. Mutational analysis of this interface across three distinct MAGEs (MAGE-A3, -A11, and -F1) established that this region defines the substrate binding site on MAGEs. Thus, we refer to this region as the substrate binding cleft (SBC). Notably, in the previously solved structures of MAGE-A3 and -A4, the SBC was occupied by unstructured intermolecular MAGE protein or tag residues, respectively[9]. Specifically, MAGE-A3 was observed as a domain-swapped dimer where the C-terminus was observed as a random coil threaded through the SBC of the crystallographic symmetry related protomer. However, in the MAGE-A11:PCF11 structure, these residues form the

terminal helix of a helical bundle structure at the C-terminus that partly contributes to substrate binding. In MAGE-A4, the N-terminal purification tag threads through the SBC as part of a domain-swapped symmetry related dimer[9]. Our results suggest that interaction of the unstructured C-termini of MAGE-A3 and N-terminal tag of MAGE-A4 with the SBC may be induced during crystallization due to the absence of a natural substrate. This could lead to the hydrophobic cleft being filled unnaturally to allow protein stabilization and crystallization. In either case, the SBC appears to be an important binding site for protein–protein interaction, including substrates.

The MAGE family is evolutionarily conserved in all eukaryotes and underwent dynamic expansion from a single gene in protozoa to a large multigene family in eutherian mammals[2]. Evolutionarily, type II MAGEs appeared earlier than type I MAGEs, but the MHD is a common feature to both type I and II MAGEs. MHDs are highly conserved (46% identical) in human, especially, the MHDs of MAGE-A subfamily share >80% protein sequence identity[7,8]. However, distinct MHDs often recognize non-overlapping substrates[5,10,15]. We found that this substrate selectivity is likely imparted by sequence diversity around the SBC that may provide key positive and negative filtering. In addition to binding substrates, the MHD also mediates binding to RING or HECT ligases. For example, MAGE-C2 MHD bound the coiled-coil region on TRIM28, MAGE-B18 bound a basic region of LNX1 and MAGE-G1 bound the WH-A motif of NSE1[8]. Although the MHD mediates ligase binding, we found that mutation of MAGE-A3, -A11, or -F1 SBC did not disrupt binding to their cognate E3 ligases. This suggests that: (1) SBC mutations do not cause global structural changes to the MHD and (2) ligase and substrate binding to the MHD occur independently on non-overlapping surfaces. Consistent with this observation, our previous co-crystal structure of MAGE-G1:NSE1 showed a distinct binding surface for NSE1 not restricted to the MHD SBC[8]. Notably, the relative orientation of WH-A and WH-B of MAGE-G1 MHD in the MAGE-G1:NSE1 structure is quite distinct compared to the structures of MAGE-A3, -A4, and -A11:PCF11[5]. Therefore, E3 binding may induce structural dynamics within the MHD and the relative positioning of the WH motifs may contribute to selective ligase binding between MAGEs. Additional MHD structures with both ligase and substrate bound will provide important answers to these questions.

Finally, MAGEs and other cancer-testis antigens have been extensively proposed as cancer-specific therapeutic targets. Thus far there has been limited success. In this study we focus on targeting oncogenic MAGEs through blocking binding to substrates. We propose this approach will be of broad utility against the many cancer-specific MAGEs. As proof-of-principle we target MAGE-A11 and discover sub-micromolar 4-Aminoquinolines inhibitors of MAGE-A11 binding to PCF11. These tool compounds will help facilitate research into MAGE-A11 biology and give precedence for the development of future MAGE-directed therapeutics.

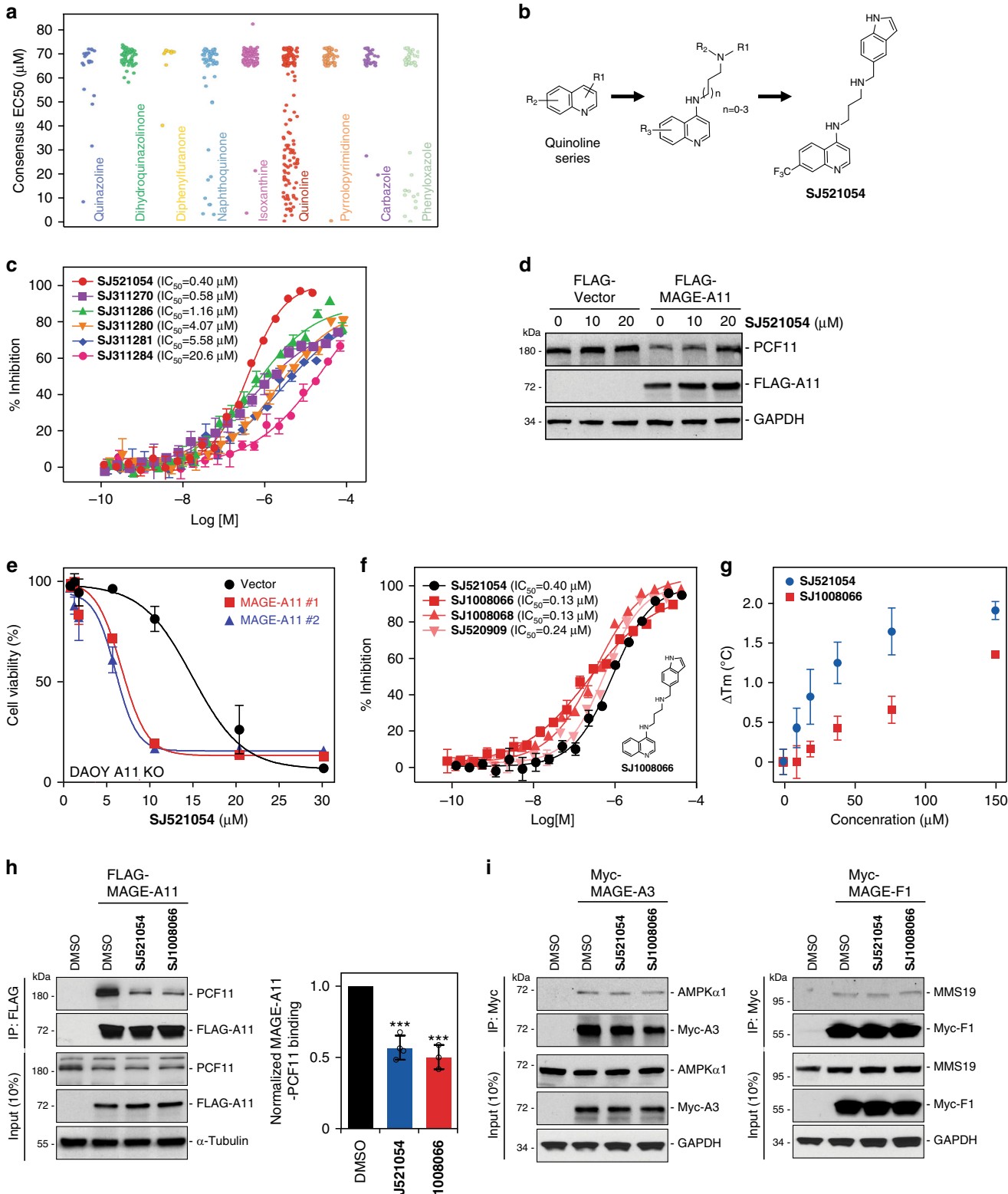

## Methods

**Cell culture and stable cell lines**. HEK293FT, HeLa and DAOY cells were grown in DMEM (Gibco) supplemented with 10% (v/v) FBS (Hyclone), 2 mM L-gluta-mine, 100 units/ml penicillin, 100 units/ml streptomycin, and 0.25 mg/ml amphotericin B (Invitrogen). HEK293FT and DAOY cells were transfected with either HA-FLAG-vector, FLAG-MAGE-A11 wild-type or SBC mutants using Effectene (QIAGEN) according to the manufacturer's protocol in 6 cm² plates. After 48 h, cells were selected with 1 μg/ml puromycin (Sigma) over 2 weeks.

**Immunoprecipitation**. HEK293FT cells were plated in 6 cm² plates and transfected 24 h later with Effectene (QIAGEN) according to the manufacturer's protocol. After 48 h, cells were washed and scraped in cold PBS, spun down, and resus-pended in lysis buffer (50 mM Tris pH 7.4, 150 mM NaCl, 1 mM DTT, 0.1% (v/v) Triton X-100, 10 mM N-Ethylmaleimide, and 1X protease inhibitor cocktail (Sigma)). Cell lysates were incubated with appropriate antibodies overnight at 4 °C and then with protein A beads for 2 h at 4 °C. Beads were then washed with lysis buffer three times and eluted with 2X SDS sample buffer.

**Fig. 5 Discovery of small molecule inhibitors targeting MAGE-A11. a** 500 analogs from 9 scaffolds were tested in dose response TR-FRET assays for disruption of MAGE-A11 binding to PCF11 identified the quinoline scaffold. **b** Chemical structure of **SJ521054** hit. **c** TR-FRET competition assay using indicated concentrations of quinoline series compounds for disruption of MAGE-A11:PCF11 binding with $n = 2$ biologically independent experiments with triplicates per compound. Data are mean ± SD. **d** **SJ521054** blocks PCF11 degradation by MAGE-A11. HEK293FT cells stably expressing FLAG-vector or FLAG-MAGE-A11 were treated with indicated concentrations of **SJ521054** for 24 h before immunoblotting. Source data are provided as a Source Data file. **e** **SJ521054** reduces viability of MAGE-A11 expressing DAOY cancer cells. MAGE-A11-knockout DAOY cells or those reconstituted with MAGE-A11 were treated with indicated concentrations of **SJ521054** and cell viability was measured by alamarBlue assay 24 h later ($n = 3$ biologically independent experiments per group). Data are mean ± SD. Black circle, Vector; Red square, MAGE-A11 WT #1; and Blue triangle, MAGE-A11 WT #2. **f** Derivatives of **SJ521054** were synthesized and tested by TR-FRET competition assays with **SJ1008066** showing improved activity ($n = 2$ biologically independent experiments with triplicates per compound). Data are mean ± SD. **g** Direct ligand binding of **SJ521054** and **SJ1008066** were evaluated using NanoDSF using the intrinsic fluorescence of His-SUMO-MAGE-A11 MHD at 350 nm and 330 nm with $n = 3$ biologically independent experiments per compound. Both compounds showed stabilization of MAGE-A11 in a six-point dose response confirming on-target activity of each compound. Data are mean ± SD. Blue circle, **SJ521054** and red square, **SJ1008066**. **h** **SJ521054** and **SJ1008066** inhibit MAGE-A11 binding to PCF11 in cell lysate. FLAG-MAGE-A11 stably expressing HEK293FT cell lysates were incubated with 100 μM of each compound for 24 h and then subjected to pull-down with anti-FLAG followed by SDS-PAGE and immunoblotting for endogenous PCF11, and quantitated ($n = 4$ for **SJ521054**, $n = 3$ for **SJ1008066**, biologically independent experiments per compound, two-tailed unpaired Student's $t$ test, ***$P < 0.001$). Data are mean ± SEM. Source data are provided as a Source Data file. **i** **SJ521054** and **SJ1008066** do not disrupt interaction of MAGE-A3:AMPKα1 or MAGE-F1:MMS19 in cell lysate. HeLa cells or HEK293FT cells were transfected with the indicated constructs for 48 h and cell lysates were incubated with 100 μM of each compound for 24 h and then subjected to IP with anti-Myc followed by SDS-PAGE and immunoblotting for indicated proteins. Source data are provided as a Source Data file.

**Western blotting and antibodies**. For western blotting, samples in SDS sample buffer were resolved on SDS-PAGE gels and then transferred to nitrocellulose membranes prior to blocking in TBST with 5% (w/v) milk powder or 3% (w/v) bovine serum albumin and probing with primary and secondary antibodies (GE Healthcare). Primary antibodies were: anti-Myc (Sigma, #C3956), anti-FLAG (Sigma, #F3165), anti-HUWE1 (Novus Biologicals, #NB 100-652), anti-PCF11 (Bethyl Laboratories, #A303-706A), anti-GAPDH (Cell Signaling Technology, #2118), anti-TRIM28 (Abcam, #ab22553), anti-AMPKα1 (Cell Signaling Technology, #2795), and anti-MMS19 (Proteintech, #16015-1-AP) and Tb-anti-GST (Invitrogen, #PV3551). Secondary antibodies were: Donkey Anti-Rabbit IgG (GE Healthcare, NA934V) and Sheep Anti-Mouse IgG (GE Healthcare, NA931V). FLAG and TRIM28 primary antibodies were used at a dilution of 1:10,000 and Tb-anti-GST antibody was used at a dilution 1:720 and other primary antibodies were used at a dilution of 1:1000. All GE Healthcare secondary antibodies were used at a dilution of 1:5000. Protein signal was visualized after addition of ECL detection reagent (GE Healthcare) according to manufacturer's instructions.

**Peptide synthesis**. N,N-dimethylformamide (DMF), N-methylpyrrolidone (NMP), trifluoroacetic acid (TFA), 2-(6-Chloro-1-H-benzotriazole-1-yl)-1,1,3,3-tetramethylaminium hexafluorophosphate (HCTU), and N-methylmorpholine (NMM), were purchased from GyrosProteinTechnologies (Tucson, AZ). O-(7-Azabenzotriazol-1-yl)-N,N,N′,N′-tetramethyluronium hexafluorophosphate (HATU), 1-Hydroxy-7-azabenzotriazole (HOAt), and substituted FMOC-phenylalanine analogs were purchased from ChemImpex (Wood Dale, IL). Piperidine, N,N-Diisopropylethylamine (DIPEA), acetic anhydride, thioanisole, triisopropylsilane (TIS), phenol, 1,2-ethanedithiol (EDT), Fmoc-Lys(Ac)-OH, diethyl ether, and Hoveyda-Grubbs 2nd generation catalyst were purchased from Sigma-Aldrich (St Louis, MO). Cyanine5 NHS ester was purchased from Lumiprobe (Hunt Valley, MD). Fmoc-(S)-2-(4-pentenyl)alanine and Fmoc-(R)-2-(4-pentenyl)alanine were purchased from Aapptec (Louisville, KY). All standard Fmoc-amino acids and preloaded Wang resins were purchased from Millipore-Sigma (Burlington, MA). Dichloromethane (DCM) was purchased from Fisher Scientific (Waltham, MA). N,N-Diisopropylcarbodiimide (DIC) and fluorescein-5-carboxylic acid (5-FAM) were purchased from Anaspec (Fremont, CA).

For standard peptide synthesis, peptides were synthesized at a 25-μmol scale on a SymphonyX peptide synthesizer (GyrosProteinTechnologies, Tucson, AZ) using preloaded Wang resins. Deprotection was performed with 20% piperidine in NMP for 3 × 5 min at room temperature. Coupling was performed with 0.2 M amino acid with a ratio of 1/5/5 amino acid/HCTU/NMM in NMP for 2 × 60 min at room temperature. Capping was performed after each amino acid cycle using 5% acetic anhydride in NMP at room temperature. After synthesis, peptides were cleaved from the resin using 82.5/5/5/2.5/2.5/2.5 -TFA/water/thioanisole/TIS/phenol/EDT for 2 h at room temperature. After filtration peptides were precipitated using ice-cold diethyl ether followed by centrifugation. Peptides were dissolved in water and lyophilized on a Freezemobile (SP Scientific, Warminster, PA). Peptides were HPLC purified using a Waters Preparative HPLC system (Waters, Milford, MA). Peptide purity was checked using an Analytical HPLC system (Waters, Milford, MA). All peptides synthesized had the expected molecular weights and were analyzed using a MicroFlex MALDI mass spectrophotometer (Bruker, Billerica, MA).

For peptides labeled with 5-FAM, peptides were synthesized at a 25-μmol scale in the same way as described above. After the synthesis was complete, the peptide resin was treated with 5-FAM/HOAt/DIC in the ratio 1/3.25/4.20 in NMP at room

temperature overnight. Peptides were cleaved, purified and analyzed as described above.

For library peptide synthesis, a peptide library of 250 peptides was synthesized at a 10-μmol scale on an Overture robotic peptide synthesizer (GyrosProteinTechnologies, Tucson, AZ) using preloaded Wang resins. Deprotection was performed with 20% piperidine in NMP for 3 × 5 min at room temperature. Coupling was performed with 0.2 M amino acid with a ratio of 1/5/5 amino acid/HCTU/NMM in NMP for 2 × 60 min at room temperature. Capping was performed after each amino acid cycle using 5% acetic anhydride in NMM at room temperature. After synthesis, peptides were cleaved from the resin using 82.5/5/5/2.5/2.5/2.5 -TFA/water/thioanisole/TIS /phenol/EDT for 2 h at room temperature. After filtration peptides were precipitated using ice-cold diethyl ether followed by centrifugation. All peptides synthesized had the expected molecular weights and were analyzed using a MicroFlex MALDI mass spectrophotometer (Bruker, Billerica, MA).

For synthesis of peptides containing modified phenylalanine, peptides were synthesized at a 25-μmol scale in the same way as described above. The modified phenylalanines were introduced by manually activating the corresponding Fmoc-phenylalanine derivatives and coupling using a ratio of 1/5/5 amino acid/HATU/ DIPEA in NMP for 2 × 60 min at room temperature. Peptides were cleaved, purified and analyzed as described above.

**TR-FRET assays and high throughput screen**. Terbium-anti-GST antibody, 1 M Tris pH 7.5 1 M MgCl$_2$ and 1 M DTT were purchased from Invitrogen (Carlsbad, CA), BSA was purchased from Sigma (St. Louis, MO). Dimethyl sulfoxide (DMSO) was purchased from Fisher Scientific (Pittsburgh, PA). Black 384-well low volume plates and 384-well polypropylene compound plates were purchased from Corning Incorporated (Tewksbury, MA). The St. Jude FDA drug and Bioactive library (11,293 chemicals), Drug-like library (10,073 chemicals), and Lead-like library (10,041 chemicals) were custom assembled and purchased from commercial sources including Sigma, Microsource (Gaylordsville, CT), selleckchem (Houston, TX), ChemDiv (San Diego, CA), ChemBridge (San Diego, CA), Enamine (Monmouth Jct., NJ), and Life Chemicals (Niagara-on-the-Lake, ON, Canada). They were arrayed (10 mM in DMSO) in 384-well polypropylene compound plates with columns 1, 2, 13, and 14 empty which were reserved for controls.

The TR-FRET assay buffer with a formula of 50 mM Tris pH 7.5, 20 mM MgCl$_2$, 0.1 mg/ml BSA and 1 mM DTT was freshly prepared before each experiment. All TR-FRET assays were performed in black 384-well plates with 20 μl/well assay volume in the TR-FRET assay buffer at room temperature (~25 °C). All peptides or compounds were solubilized in DMSO. To test the competitive activity of peptides or compounds against the interaction between GST-MAGE-A11 MHD and FAM-PCF11_4 peptide, 10 μl/well peptide or compound at specified concentration was dispensed into a black 384-well low volume plate, followed by 5 μl/well FAM-PCF11_4 (800 nM), 5 μl/well Tb-anti-GST (20 nM), and GST-MAGE-A11 MHD (20 nM). The assay components in each well was then mixed by shaking the plate on an IKA MTS 2/4 digital microtiter plate shaker (IKA Works; Wilmington, NC, USA) for 1 min. The plate was further centrifuged in an Eppendorf 5810 centrifuge with an A-4-62 swing-bucket rotor (Eppendorf AG, Hamburg, Germany) at 201 g for 30 s. The plate was then incubated for 90 min with a lid to avoid light exposure to the assay components. After incubation, the fluorescent emission signals at 520 nm and 490 nm channels of individual wells were measured with a PHERAstar FS plate reader (BMG Labtech; Durham, NC, USA) by using a 340 nm excitation filter, 100 μs delay time,

and 200 μs integration time. The TR-FRET fluorescence emission ratio (TR-FRET signal) for each well was presented as 10,000 × 520 nm/490 nm, which was used directly for curve fitting, or further converted to %Inhibition based on the respective positive and negative controls. The graphic software GraphPad Prism 4.81 (GraphPad Software; La Jolla, CA, USA) was used to generate curves and derive IC$_{50}$ values of tested peptides or compounds, if applicable.

TR-FRET inhibitory activity characterization of a panel of PCF11 peptides against the interaction between GST-MAGE-A11 MHD and FAM-PCF11_4 was performed as follows. In a 20-μl/well assay volume, dilutions of PCF11_1 to PCF11_8 peptides (concentrations ranged from 3.1 nM to 100 μM for peptides PCF11_4, PCF11_5, PCF11_6 and PCF11_8 in a 1-to-2 dilution pattern for 16 concentration levels, and ranged from 48.9 nM to 100 μM for peptides of PCF11_1, PCF11_2, PCF11_3, and PCF11_7 in a 1-to-2 dilution pattern for 12 concentration levels), 200 nM FAM-PCF11_4, 5 nM Tb-anti-GST and 5 nM GST-MAGE-A11 were mixed together and incubated for 90 min before the TR-FRET signal for each well was measured. The final DMSO concentration was 1.1% (v/v). In each group, a group of wells with 1.1% (v/v) DMSO along with 200 nM FAM-PCF11_4, 5 nM Tb-anti-GST, and 5 nM GST-MAGE-A11 MHD (DMSO group) was included as the negative control (Signal$_{DMSO}$, 0% inhibition). A group of wells with 1.1% (v/v) DMSO along with 200 nM FAM-PCF11_4 and 5 nM Tb-anti-GST (DMSO without GST-MAGE-A11 group) was also included as the positive control (Signal$_{DMSO without GST-MAGE-A11}$, 100% inhibition). The signal of each peptide-treated well (Signal$_{peptide}$) was normalized to the positive and negative controls to derive %inhibition of the peptide at a specific concentration with Eq. (1). Where applicable, the %Inhibition data were fit into a Sigmoidal dose response equation to derive IC$_{50}$ values.

$$\%\text{Inhibition} = 100\% - 100\% \times \frac{(\text{Signal}_{peptide} - \text{Signal}_{DMSO\,without\,GST-MAGEA11})}{(\text{Signal}_{DMSO} - \text{Signal}_{DMSO\,without\,GST-MAGEA11})}. \quad (1)$$

TR-FRET screen for identifying MAGE-A11 small molecule inhibitors was performed as follows. The primary one-concentration screening and subsequent 10 or 20-concentration dose-response tests were accomplished with an automated High Throughput Screening system (HighRes Biosolutions, Beverly, MA)[30]. In the primary screening, 15 μl/well FAM-PCF11_4 (267 nM) was first dispensed with a Multidrop Combi Reagent Dispenser (Thermo Fisher Scientific, Waltham, MA) to black 384-well low-volume assay plates. The plates were spun down with a Velocity11 VSpin microplate centrifuge (Agilent Technologies, Santa Clara, CA). In total, 30 nl/well of 10 mM compound/DMSO solutions, or DMSO alone, were transferred from compound plates (the St. Jude FDA drug and bioactive library, Drug-like library and Lead-like library) or a control plate to the assay plates with a Pin Tool (V&P Scientific, San Diego, CA). The compounds were arrayed in columns 3–12 and columns 15–24 of compound plates. The controls were arrayed in columns 1, 2, 13, and 14 of a compound plate. In all, 5 μl/well Tb-anti-GST (20 nM) and GST-MAGE-A11 (20 nM) or 5 μl/well Tb-anti-GST (20 nM) alone was then dispensed to selected wells with a Multidrop Combi Reagent Dispenser. The final assay volume was 20 μl/well. The final DMSO concentration was 0.25% (v/v) (0.1% (v/v) from the FAM-PCF11_4 DMSO stock solution and 0.15% (v/v) from the compound-DMSO solution or the DMSO for the control wells). The final tested compound concentration was 15 μM. The final FAM-PCF11_4 concentration was 200 nM and the final GST-anti-GST concentration was 5 nM. The final GST-MAGE-A11 concentration was 5 nM except for those selected positive control wells in which GST-MAGE-A11 was not presented. The plates were then shaken with a big bear shaker (Big Bear Automation, Santa Clara, California) and spun down with the Velocity11 VSpin microplate centrifuge for 20 s. The plates were then incubated at room temperature in a Liconic incubator (Liconic US, Woburn, MA) for 90 min. The TR-FRET signals for individual wells were measured with a PHERAstar FS plate reader. The %Inhibition of each tested inhibitor was calculated by using Eq. 1. Those inhibitors with %Inhibition >30% in the primary screening were selected for further dose-response test, along with additional newly synthesized compounds.

The Z′-factor of each screening plate was calculated by using Eq. (2)[27] with 16 data points in both the negative control (the DMSO group) and the positive control (DMSO without GST-MAGE-A11 group) included.

$$Z' = 1 - \frac{3\sigma^+ + 3\sigma^-}{\text{Mean}^+ - \text{Mean}^-}. \quad (2)$$

where $\sigma+$ is the standard deviation of the negative control group (DMSO group), $\sigma-$ is the standard deviation of the positive control group (DMSO without GST-MAGE-A11 group), mean+ is the mean of the negative control group (DMSO group), and mean− is the mean of the positive control group (DMSO without GST-MAGE-A11 group).

In the TR-FRET dose response test, a protocol similar to the primary screening was followed except that the final DMSO concentration was 0.8% (v/v) with 0.1% (v/v) from the FAM-PCF11_4 DMSO stock solution and 0.7% (v/v) from the compound-DMSO solution or the DMSO for the control wells (Pin Tool was used to transfer 140 nl of compound-DMSO solutions or DMSO). The inhibitors were arrayed in columns 1–24 and the controls were arrayed in columns 21–24 of 384-well compound plates. The final tested compound concentrations ranged from 3.6 nM to 70 μM in a 1-to-3 dilution pattern for 10 concentration levels or ranged

from 0.13 nM to 70 μM in a 1-to-2 dilution pattern for 20 concentration levels. The %Inhibitions of each tested inhibitor at individual concentrations were calculated by using Eq. 1 and were fit into a Sigmoidal dose response equation to derive IC$_{50}$ values with the GraphPad Prism 4.81 software, if applicable.

**Thermal stability measurements.** The thermal stability of the MAGE-A11 in the presence of ligand was measured using differential scanning fluorimetry. In all, 200 μM of ligand PCF11_6, diluted to achieve the desired final concentration in DMSO, was transferred to an Applied Biosystems MicroAmp Optical reaction plate using acoustic transfer. To each well, 20 μl thermal stability assay solution (10 μM His-SUMO-MAGE-A11 and 5x SyproOrange in 50 mM HEPES, 100 mM NaCl, 50 mM MgCl$_2$, pH 7.5) was added, and the plate was sealed with MicroAmp optical adhesive film. The plate was exposed to a range of temperatures (25–99 °C, ramp rate 0.1 °C/s) while measuring the fluorescence of SyproOrange in an Applied Biosystems QuantStudio 5 real-time PCR instrument. Resulting data were then analyzed using GraphPad Prism (v8.3.1), and the melting temperatures of each condition calculated using the Boltzmann Equation. All experiments were performed in triplicate.

**Nano-differential scanning fluorimetry measurements.** For NanoDSF measurements, a Prometheus NT.48 instrument (NanoTemper Technologies) was used to determine the melting temperatures. Triplicate samples were prepared in a standard 384-well PCR plate with His-SUMO-MAGE-A11 at 10 μM and ligand concentrations as noted. After incubating the protein and ligands for at least 30 min, capillaries were filled with 10 μL of sample and placed on the sample holder. DMSO was used as a negative reference control while PCF11_6 peptide was used as a positive control validating the assay. A temperature gradient of 1 °C min$^{-1}$ from 25 to 95 °C was applied and the intrinsic protein fluorescence at 330 and 350 nm was recorded. The 350 nm to 330 nm ratios of fluorescence measurements were used to calculate the $T_m$ values of each sample using the Nanotemper PR.ThermControl software. Data were further analyzed and plotted using GraphPad Prism (v 8.3.1).

**Protein expression and purification.** GST-PCF11 (full-length *human* PCF11 or amino acids 637-702), GST-MAGE-A11 MHD (*human* MAGE-A11 MHD amino acid 218-429), or GST tag alone were induced in BL21(DE3) cells at 16 °C with 0.5 mM isopropyl β-D-1-thiogalactopyranoside (IPTG). GST-tagged proteins were purified from bacterial lysates in lysis buffer (50 mM Tris pH 7.7, 150 mM KCl, 0.1 % (v/v) Triton X-100, 1 mM DTT, 1 mg/ml lysozyme) using GSTrap Hp 5 ml column (GE Amersham) followed by Resource Q ion exchange chromatography (GE Amersham) and was concentrated to 15 mg/ml in 20 mM Tris pH 7.4, 150 mM NaCl, 1 mM DTT and 5% (v/v) glycerol.

For crystallization, PCF11 peptide (*human* PCF11 amino acid 677-701) is fused at the N-terminus of MAGE-A11 MHD (218-429) with linker sequence GGSGRP. The fusion protein was expressed in BL21(DE3) *E. coli* using the pSUMO3 vector. The protein complex was purified using a HisTrap Hp 5 ml column (GE Amersham) followed by Resource Q ion exchange chromatography (GE Amersham). The His-SUMO tag was subsequently cleaved off by addition of SENP protease followed by secondary HisTrap Hp 5 ml column (GE Amersham). The resulting fusion protein complex was concentrated to 15 mg/ml in 50 mM Tris, pH 7.4, 200 mM NaCl, and 5 mM DTT.

**Crystallization, data collection, structure determination, and model quality.** Crystals were grown by the sitting-drop vapor diffusion method at 20 °C. The well solution contained 0.2 M Na acetate, 0.1 M Bis Tris propane pH 7.5 and 20% (w/v) PEG 3350. The drop contained a 1:1 volume ratio of well solution to protein (15 mg/ml in 50 mM Tris, pH 7.4, 200 mM NaCl and 5 mM DTT). Crystals appeared overnight and matured in ~1 week. For cryo-preservation, the crystals were incubated in reservoir solution with 10% (v/v) glycerol prior to flash-cooling in liquid nitrogen. Data were collected at the Southeast Regional Collaborative Access Team (SER-CAT) Sector 22-BM beamline at 1 Å. Data were integrated and scaled using HKL2000 v716[31] to 2.2 Å. The structure was solved by molecular replacement with Phaser v2.8[32] using MAGE-A3 (PDB ID 4V0P) as the search model. Iterative rounds of model building and refinement were performed with COOT v0.8.9[33] and Refmac v5.8.0257[34], respectively. Processing and refinement statistics are summarized in Table 1.

The final model is of high quality, with four fusion protein protomers in the P 1 cell. Although PCF11 peptide residues were readily identified in all protomers, the MAGE-A11 linker was disordered. All protomers are remarkably similar, with a C-alpha RMSD of <0.4 Å relative to protomer A (chain B 0.141, chain C 0.351, and chain D 0.349). Detailed descriptions and figures refer to protomer A, where PCF11 is solvent exposed and unperturbed by crystal packing. The coordinates and structure factors have been deposited in the Protein Data Bank with accession number 6WJH. Figures were made using PyMOL[35].

**GST-pulldown in vitro binding assays.** Myc-tagged proteins were in vitro translated using the SP6-TNT Quick rabbit reticulocyte lysate system (Promega) according to manufacturer's instructions. In vitro binding assays were performed by incubating 15 mg of purified GST-tagged protein PCF11 (*human* PCF11 amino acid 637-702) or MAGE-A11 (*human* MAGE-A11 amino acid 218-429) with

glutathione Sepharose beads (GE, Amersham) for 1 h in binding buffer (25 mM Tris pH 8.0, 2.7 mM KCl, 137 mM NaCl, 0.05% (v/v) Tween-20, and 10 mM 2-mercaptoethanol). Bound beads were blocked for 1 h in binding buffer containing 5% (w/v) milk powder. In vitro translated proteins (5 µl; Promega TNT rabbit reticulocyte lysate quick SP6 system) were then incubated with the bound beads for 1 h in binding buffer containing 5% (w/v) milk powder. After four washes in binding buffer, the proteins were eluted in SDS-sample buffer, boiled, subjected to SDS-PAGE, and blotted with anti-Myc.

**Tandem ubiquitin binding entity ubiquitination assay**. One 10 cm$^2$ plates of HEK293FT cells stably expressing FLAG-vector, FLAG-MAGE-A11 wild-type or SBC mutants were lysed with tandem ubiquitin binding entity (TUBE) lysis buffer (50 mM Tris pH 7.5, 150 mM NaCl, 1 mM EDTA, 1% (v/v) NP-40, 10% (v/v) glycerol, 20 mM *N*-Ethylmaleimide and 1X protease inhibitor cocktail), and the lysates were bound to TUBE-agarose (LifeSensors) overnight at 4 °C. Beads were subsequently washed three times in wash buffer (20 mM Tris pH 8.0, 150 mM NaCl, and 0.1% (v/v) Tween 20) and then the ubiquitinated proteins were eluted in SDS sample buffer, boiled, and subjected to SDS-PAGE and western blotting.

**Clonogenic growth assay**. For clonogenic growth assays on plates, wild-type MAGE-A11 or SBC mutants reconstituted MAGE-A11-knockout DAOY cells were plated in six-well plates in triplicate. After 2–3 weeks, cells were fixed and stained with 0.05% (w/v) crystal violet and counted (colonies ≥100 µm in size).

**Cell viability assay**. To assess cell viability after **SJ521054** treatment, $1 \times 10^4$ cells/ml were treated with **SJ521054** (0–40 µM) and incubated for 24 h prior to changing the media and adding alamarBlue (Thermo Fisher Scientific) and incubating for 4 h at 37 °C. Plates were read by measuring the fluorescence with excitation wavelength at 540 nm and emission wavelength at 590 nm on an Enspire plate reader.

**Xenograft tumor growth assays**. In all, $3 \times 10^6$ MAGE-A11-knockout DAOY cells reconstituted with wild-type MAGE-A11 or SBC mutants were mixed with matrigel (Corning) before injection into the flank of NOD scid gamma mice (Jackson Lab; $n = 6$ for each group). Tumor size was measured 2–3 times a week during the duration of the experiment. All animal experiments were approved by the St. Jude Children's Research Hospital's IACUC.

**RNA-seq and 3′-UTR analysis**. Total RNA was extracted from xenograft tumors using RNeasy kit (Qiagen) according to manufacturer's instructions. RNA quality was assessed by 2100 Bioanalyzer RNA 6000 Nano assay (Agilent). Libraries were prepared using TruSeq Stranded mRNA kits (Illumina) and subjected to 100 cycle paired-end sequencing on the Illumina HiSeq platform. Low quality reads were filtered out using Trim Galore and then aligned to the human genome (hg19/GRCh37) using STAR version 2.5.2b[36]. The mapped BAM files were converted into bedgraph format using bedtools version 2.17.0[37]. The RNA-seq read coverage was visualized at UCSC Genome Browser[38]. Data was deposited to NCBI gene expression omnibus as accession GSE148458.

DaPars[39] was used to identify the most significant APA events between MAGE-A11 WT and M341R conditions. RNA-seq coverage for 3′UTR region had to be >20 to be included for analysis. The significant APA events were defined as (1) the adjusted *P* value was controlled at 5%. (2) the absolute mean difference of PDUI must be no less than 0.2. (3) the mean PDUI fold change must be no <1.5.

**Chemical synthesis**. Typical synthetic routes used to explore the structure–activity relationship of quinoline analogs are shown in Supplementary Fig. 6. Detail protocols for the synthesis of quinoline analogs are described in full in the supplementary information methods and Supplementary Fig. 6[40–42].

**Small molecule solubility and permeability studies on small molecules**. Solubility assays were carried out on a Biomek FX lab automation workstation (Beckman Coulter, Inc., Fullerton, CA) using µSOL Evolution software (pION Inc., Woburn, MA). The detailed method is described as follows. 10 µl of 10 mM compound stock (in DMSO) was added to 190 µl 1-propanol to make a reference stock plate. In total, 5 µl from this reference stock plate were mixed with 70 µl 1-propanol and 75 µl citrate phosphate-buffered saline (PBS; isotonic) to make the reference plate, and the UV spectrum (250–500 nm) of the reference plate was read. In total, 6 µl of 10 mM test compound stock was added to 594 µl buffer in a 96-well storage plate and mixed. The storage plate was sealed and incubated at RT for 18 h. The suspension was then filtered through a 96-well filter plate (pION Inc., Woburn, MA). In all, 75 µl of filtrate was mixed with 75 µl 1-propanol to make the sample plate, and the UV spectrum of the sample plate was read. Calculations were carried out with µSOL Evolution software based on the area under the curve (AUC) of the UV spectrum of the sample and reference plates. All compounds were tested in triplicate.

High throughput Caco-2 permeability was performed in the TranswellR 0.4 µm polycarbonate membrane 96-well system with modified methods. Caco-2 cells were maintained at 37 °C in a humidified incubator with an atmosphere of 5% $CO_2$. The cells were cultured in 75 cm$^2$ flasks with Dulbecco's Modified Eagle's Medium

(DMEM) containing 10% (v/v) FBS, 1% (v/v) non-essential amino acids (NEAA), 100 units/ml of penicillin, and 100 µg/ml of streptomycin. The Caco-2 cells were seeded onto inserts at a density of $2 \times 10^4$ cells/insert separately. The medium in the wells was exchanged each other day, and the trans epithelial electrical resistance (TEER) value was measured using an epithelial voltohm meter (Millipore, Billerica, Massachusetts). Caco2 cells were grown for 7 days to reach consistent TEER values (typically 2000 ohms greater than initial value when cells are first seeded into transwells), indicating that the cells had formed a confluent polarized monolayer. For transport experiments, each cultured monolayer on the 96-well plate was washed twice with a transport buffer (HBSS containing 25 mM HEPES, pH 7.4). The permeability assay was initiated by the addition of each compound solution (10 µmol/l) into inserts (apical side, A) or receivers (basolateral side, B). The Caco-2 cell monolayers were incubated for 2 h at 37 °C. Fractions were collected from receivers (if apical to basal permeability) or inserts (if basal to apical permeability), and concentrations were assessed by UPLC/MS (Waters; Milford, MA). All compounds were tested in triplicates. The A → B (or B → A) apparent permeability coefficients (Papp) of each compound were calculated using the equation, Papp = dQ/dt × 1/AC0, where dQ/dt equals the flux of a drug across the monolayer, A equals the total insert well surface area, and C0 is the initial concentration of substrate in the donor compartment. The efflux ratio was determined by dividing the Papp in the B-A direction by the Papp in the A-B direction. An efflux ratio >2 suggested that a given substrate was actively transported across the membrane.

cLog P and tPSA values provided in the Supplementary Data 4 were estimated using the PerkinElmer ChemDraw Professional, version 17.1.0.105 software[43].

**Statistics and reproducibility**. Data shown in graphs represent mean ± standard error of the mean (SEM) or mean ± standard deviation (SD), as indicated in the figure legends. Statistical analysis was performed at least three independent experiments and analyzed with GraphPad Prism 7. Statistical significance was analyzed using unpaired Student's *t* test, ordinary one-way or two-way ANOVA with Sidak's multiple comparisons test for multiple groups. For western blotting analysis, each experiment was repeated at least two times independently with similar results. Each experiment was repeated: Fig. 1a ($n = 3$), h ($n = 2$); 2a ($n = 3$), e ($n = 2$), f ($n = 3$); 3a ($n = 3$); 4c ($n = 2$), e ($n = 3$), g ($n = 3$); and 5d ($n = 3$); Supplementary Fig. 2a ($n = 3$) and 4e ($n = 3$).

**Reporting summary**. Further information on research design is available in the Nature Research Reporting Summary linked to this article.

## Data availability

The datasets generated during and/or analyzed during the current study are available from the corresponding author on reasonable request. Structural data that support the findings of this study have been deposited in PDB with the accession code PDB: 6WJH. Sequencing data that support the findings of this study have been deposited in the GEO with the accession code GSE148458. Source data are provided with this paper.

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

## Acknowledgements

We thank members of the Potts lab for helpful discussions and critical reading of the manuscript. We also thank the St. Jude X-Ray Center, Hartwell Center and Department of Chemical Biology and Therapeutics at St. Jude. Use of the Advanced Photon Source was supported by the U.S. Department of Energy, Office of Science, Office of Basic Energy Sciences, under Contract No. W-31-109-Eng-38. Data were collected at Southeast Regional Collaborative Access Team (SER-CAT) 22-BM beamline at the Advanced Photon Source, Argonne National Laboratory. SER-CAT is supported by its member institutions (see www.ser-cat.org/members.html), and equipment grants (S10_RR25528 and S10_RR028976) from the National Institutes of Health. This work was partially supported by American Cancer Society Research Scholar Award 181691010 (P.R.P.), a sponsored research agreement from Amgen (P.R.P.) and NIH/NCI R01CA193466 (W.L.).

## Author contributions

P.R.P., S.W.Y. and X.H. conceptualized the study and designed experiments. X.H. and S.W.Y. performed protein based experiments with D.J.M., W.L., J.M., P.R., E.C.G. and C.T.G. X.H. and D.J.M. designed and executed crystallographic studies. W.Lin. developed TR-FRET assays and performed peptide and chemical screens. S.W.Y. performed cell based and animal work. J.M. and A.M. performed chemical analysis and synthesis. P.R. performed peptide synthesis. E.C.G. and C.T.G. performed thermal stability measurements. L.L. performed RNAseq analysis. P.R.P., T.C., Z.R., R.E.L. and W.Li. supervised work and provided mentorship. All authors contributed to preparation of the manuscript.

## Competing interests

P.R.P. is a paid consultant of Levo Therapeutics, Inc. and Amgen Inc. All other authors declare no competing interests.
