## [Peer Review File · Nature Communications]

REVIEWER COMMENTS

Reviewer #1 (Remarks to the Author):

•What are the major claims of the paper? MAGE proteins bind to ubiquitin E3 ligases and facilitate ubiquitination and degradation of specific substrate proteins. This group had previously solved the structure of several MAGE proteins and identified a critical di-leucine motif that is highly conserved and is necessary for MAGE proteins to bind to and activate their specific ligases. Previous studies, however, identified only a few substrates and did not explain the specificity of MAGE/RING ligase binding of substrates. The authors now present discovery of a MAGE substrate binding cleft (SBC) that lends specificity to substrate binding, independent of the di-leucine ligase binding motif. Additionally, the authors identify potential lead compounds targeted at the SBD, the most potent of which show sub-micromolar LC50s.

Are the claims novel? There have been hints of a pocket or cleft between the MAGE winged helices (Newman JA, et al. (2016) Structures of Two Melanoma-Associated Antigens Suggest Allosteric Regulation of Effector Binding. PLoS ONE 11(2): e0148762. doi:10.1371/ journal.pone.0148762) but definitive studies of this area have not been performed and the identification of the MAGE SBC is novel and important, offering potentially tumor specific therapeutic targets.

•Will the paper be of interest to others in the field?

Yes, these studies will be of great interest to basic scientists, oncologists and scientists studying reproduction.

•Will the paper influence thinking in the field? Yes, these findings will help focus on targeting the functions of MAGE proteins by therapeutic agents rather than their use as targets for immunotherapy.

•Are the claims convincing? Yes, the authors are fastidious in their approach, verifying each step of their story with multiple different studies using complementary techniques to support their arguments. These studies that include numerous in vitro binding assays, crystallization and structure function relationship analysis, in vitro and in vivo tumorigenicity assays , TR-FRET and medicinal chemistry. The authors appear to carefully and cautiously interpret their results.

•Are there other experiments that would strengthen the paper further? A minor improvement, that could be rapidly performed without too much effort, would be to mark the conserved MAGE leucines into the molecular structures so the reader could more easily visualize the MAGE/LIGASE complex.

•Are the claims appropriately discussed in the context of previous literature? yes.

•Is the manuscript clearly written?

It is clearly written.

•Have the authors done themselves justice without overselling their claims?

Yes, this is a very important study that could eventually affect the approach to patients with many types of cancers

•Have they provided sufficient methodological detail that the experiments could be reproduced? Yes.

•Should the authors be asked to provide further data or methodological information?

No, I do not think more detail is necessary.

No ethical issues.

B. Jack Longley

Reviewer #2 (Remarks to the Author):

In this manuscript submitted by Yang, Potts (corresponding author) et al., a comprehensive, thorough, and interesting analysis of the MAGE-A11 interaction with E3 ubiquitin ligases, in particular PCF11, is presented. A combination of functional analysis of the PCF11 interaction epitope and co-crystallography were used to determine the minimal binding peptide and a 'conserved substrate binding cleft (SBC) in MAGEs'. A position-specific randomized peptide array study further supports the mutagenesis result and led to identification of a conserved peptide binding motif and discovery of several new MAGE-A11 interacting proteins. This observation was corroborated by alanine-scanning mutagenesis of PCF11 and demonstration that 'MAGE-A11 SBC mutants fail to promote proliferation of MAGE-A11 knockout DAOY cells', as well as xenograft studies. A TR-FRET assay using FAM-labeled PCF11 peptide was used for approximately 31K HTS. The HTS assay and screening data appear sound with standard statistics. However, there is lack of broad orthogonal assay testing for fluorescence artifacts and lack of solid biophysical data to demonstrate hit validation. Overall though and especially prior to the HTS section, the conclusions presented in the manuscript are well supported by data where the crystallography is particularly impressive and informative. The manuscript is well-written, presented logically, and is fairly easy to read. If published, it should be of general interest and will certainly be of interest to researchers working in the field of Melanoma Antigens and ubiquitin pathways. Addressing several questions and comments below, which mostly relate to small molecule inhibitors and 'druggability' of the MAGE protein family, should improve the manuscript:

1) Biophysical evidence, such as isothermal titration calorimetry (ITC) and/or surface plasmon resonance (SPR) (or ideally, yet more difficult, co-crystallography), that the hit compound series and preferably, SJ1008066, directly bind to MAGE-A11 with an appropriate stoichiometry (1:1) and affinity would add significant confidence to the small molecule inhibitor(s). Some orthogonal assay data is shown for SJ521054 and analogs (e.g. a pulldown experiment using 100 uM compound), but data for SJ1008066 appears missing.

2) Additional direct and selective small molecule experiments, as noted above, are especially important since the 4-aminoquinoline series, including SJ1008066 are rather 'feature-less' and might be prone to promiscuous inhibition of hydrophobically-driven protein:protein and protein:peptide interactions.

3) Above confirmatory experiments are preferred and desired for publication of the small molecule section of the manuscript. But, it is unclear why small molecule hits weren't at least crudely tested with differential scanning fluorimetry since that method was used for peptides as shown in Extended data Figure 2. Also, it would be desirable to show at least some representative examples of thermal melts in extended data.

4) Additional selectivity data for SJ1008066 amongst other MAGE family members would significantly add to the manuscript. Either result, that the small molecule inhibitor is selective or pan-MAGE, would be very interesting. It is understood that a validated small molecule inhibitor may potentially be MAGE-specific given the MAGE family discussion herein and data shown in Figure 4, however some supportive data would be an improvement.

5) Were traditional peptide co-crystal structures attempted and failed leading to the MAGE-A11:PCF11 peptide fusion construct? Including more information here would be beneficial to readers.

6) Concerning the proteome search using the consensus MAGE-A11 interacting peptide, it might be valuable to additionally search for where the peptide motif is located in the protein if a crystal structure exists. For example, the lack of confirmation that MTPN (myotrophin) interacts with MAGE-A11 is likely explained by the fact that the residues shown to be important for binding are not solvent exposed (brief analysis of PDB: 3AAA). In fact, it is often observed that linear sequence similarity to a short peptide motif does not correlate to presentation of that motif on a properly folded protein. This a rather minor point, but some additional discussion/analysis here would further support the approach taken and the observation that RACGAP1, ANKRD65 and ZBTB7C are bona fide MAGE-A11 interaction partners. Also, it is suggested that the word 'robust' be changed or removed (page 7, line 136) since only a crude pulldown experiment from IVT was completed. Definitive confirmation would require an alternate and more quantitative assay format and competition studies with PCF11 peptide.

Very minor typos/edits:

1) Comma after Finally; page 16, line 349.

- 2) 200 n of ligand PCF11_6; page 24, line 537.
- 3) Missing period: acids Notably, the PCF11 sequence; line 131, page 7.
- 4) Either change to 'agonists' (actually, think this small molecule is an 'antagonist' for IL-2 binding or add 'a' and 'inhibits': including small molecule agonist that inhibit T-cell proliferation; lines 251/252, page 12.
- 5) Double check screening metrics throughout text and figures: for example, ED Fig. 4a states '15 4-aminquinolines studied' and ED Table 5 title (Extended Data Table 5: TR-FRET and ADME results of 14 4-Aminoquinoline compounds) shows 14 compounds (text states 9 additional compounds).
- 6) Seems that 'proteins' is more appropriate here: 'the underlying function and biochemical activity of MAGE genes'; page 14, line 296
- 7) Choice of the word 'substrate' for the binding cleft is a little unusual since MAGEs are not enzymes.
- 8) ADME in section title and table title really is not appropriate for solubility and Caco-2 permeability studies.

Reviewer #3 (Remarks to the Author):

MAGE proteins serve as substrate adapters for single polypeptide RING and HECT E3 ligases thereby augmenting their substrate scope. MAGE proteins can drive cancer and this is exemplified by the protein MAGE-A11. How MAGE proteins recognise substrates is unknown and therapeutic disruption of MAGE-substrate interactions could have clinical value. The manuscript by Yang et al. identifies a substrate binding cleft within the E3 ligase adapter protein MAGE-A11. A minimal degron within the MAGE-A11 substrate PCF11 is subsequently mapped using a peptide library coupled to a TR-FRET binding assay. A general recognition motif is determined and searched against the human proteome. Disruption of the MAGE binding cleft is also shown to ablate substrate recognition in cells.

A crystal structure is solved revealing the structural basis for recognition. Finally, a HTS is carried out and proof of concept inhibitors are identified.

The assays, structural work and HTS are carried out to a high standard and on the whole the data are well interpreted.

I have the following major concerns:

"Given the importance of the C-terminal end of the PCF11 motif for MAGE-A11 binding and that MAGE-A11:PCF11 interaction affinity is not fully optimized, we synthesized several peptides with non-natural phenylalanine derivatives at PCF11 F693."

It is not entirely clear why the C-terminal end of the peptide was targeted for non-canonical amino acid mutagenesis.

What is meant by the interaction affinity not being fully optimized?

SJ521054 binding to MAGE-A11 should be quantified with a complementary biophysical technique such as ITC or SPR.

Please provide a representative blot for the quantification in figure 5d to make this more convincing. Can the authors explain/discuss why when in 5d 100 uM compound in cell lysates reduces MAGE-A11-PCF11 binding 2-fold yet in 5f, 20 uM is sufficient to reduce viability to near zero? Also in a MAGE-A11 independent manner?

The AlamarBlue assay used in 5f measures proliferation, similarly to the data presented in 3b. In 3b KO cell proliferation is reduced 2-fold relative to A11 overexpressing cells. Therefore in 5f, it would be expected that the proliferation of the vector control cells is also reduced ~2-fold relative to A11 overexpressing lines. Therefore, It is not clear if the A11 dependence is wholly due to compound binding to A11 or increased assay signal as a result of the higher proliferation rate.

Is MAGE-A11 monomeric in solution? This could be readily tested by SEC-MALS for example.

Minor concerns:

There are numerous typos throughout e.g phenylalanine is frequently incorrect.
The supplemental also has numerous typos e.g:

Page 2 Line 5: chromatograpy

Page 3 line 8, Page 5 line 22: hexanes-ethyl actate

Page 4 line 14: sodium borodydride

Page 9 line 27 and throughout: Biotage 11g NH colomn

Response to reviewers:

Reviewer #1: MAGE proteins bind to ubiquitin E3 ligases and facilitate ubiquitination and degradation of specific substrate proteins. This group had previously solved the structure of several MAGE proteins and identified a critical di-leucine motif that is highly conserved and is necessary for MAGE proteins to bind to and activate their specific ligases. Previous studies, however, identified only a few substrates and did not explain the specificity of MAGE/RING ligase binding of substrates. The authors now present discovery of a MAGE substrate binding cleft (SBC) that lends specificity to substrate binding, independent of the di-leucine ligase binding motif. Additionally, the authors identify potential lead compounds targeted at the SBD, the most potent of which show sub-micromolar LC50s.

We thank the reviewer for the kind and thoughtful comments.

Minor point: A minor improvement, that could be rapidly performed without too much effort, would be to mark the conserved MAGE leucines into the molecular structures so the reader could more easily visualize the MAGE/LIGASE complex.

The leucines are now shown in Fig. 2d (gray). They are adjacent to the SBC and part of the hydrophobic fold of the protein. We have noted in this in the text p.11-12.

Reviewer #2: In this manuscript submitted by Yang, Potts (corresponding author) et al., a comprehensive, thorough, and interesting analysis of the MAGE-A11 interaction with E3 ubiquitin ligases, in particular PCF11, is presented. A combination of functional analysis of the PCF11 interaction epitope and co-crystallography were used to determine the minimal binding peptide and a ‘conserved substrate binding cleft (SBC) in MAGEs’. A position-specific randomized peptide array study further supports the mutagenesis result and led to identification of a conserved peptide binding motif and discovery of several new MAGE-A11 interacting proteins. This observation was corroborated by alanine-scanning mutagenesis of PCF11 and demonstration that ‘MAGE-A11 SBC mutants fail to promote proliferation of MAGE-A11 knockout DAOY cells’, as well as xenograft studies. A TR-FRET assay using FAM-labeled PCF11 peptide was used for approximately 31K HTS. The HTS assay and screening data appear sound with standard statistics. However, there is lack of broad orthogonal assay testing for fluorescence artifacts and lack of solid biophysical data to demonstrate hit validation. Overall though and especially prior to the HTS section, the conclusions presented in the manuscript are well supported by data where the crystallography is particularly impressive and informative. The manuscript is well-written, presented logically, and is fairly easy to read. If published, it should be of general interest and will certainly be of interest to researchers working in the field of Melanoma Antigens and ubiquitin pathways. Addressing several questions and comments below, which mostly relate to small molecule inhibitors and ‘druggability’ of the MAGE protein family, should improve the manuscript:

We thank the reviewer for interest in our manuscript and thoughtful critiques. Below we address the concerns raised, including providing additional biophysical and experimental evidence for the proposed model.

Biophysical evidence, such as isothermal titration calorimetry (ITC) and/or surface plasmon resonance (SPR) (or ideally, yet more difficult, co-crystallography), that the hit compound series and preferably, SJ1008066, directly bind to MAGE-A11 with an appropriate stoichiometry (1:1) and affinity would add significant confidence to the small molecule inhibitor(s).

We understand the reviewer’s concern and agree that biophysical experiments would provide additional support for direct binding of small molecules to MAGE-A11. We have tried preferable approaches, such as ITC or SPR to address these issues. We faced challenges with expression of enough high quality apo-MAGE-A11 protein that precluded us from conducting ITC experiments. For SPR, His-MAGE-A11 protein appeared to load onto the chips at a sufficient level, but injection of ligands, including positive control PCF11 peptides, led to erratic sensorgrams, indicative of poor stability on the chip or interactions with the chip surface. As an alternative, we were able to perform NanoDSF to provide evidence of direct binding of both SJ521054 and SJ1008066 to MAGE-A11 MHD (Fig. 5g and Extended Data Fig. 5b,c). We believe that these data provide orthogonal validation that the small molecules directly bind MAGE-A11.

2. Some orthogonal assay data is shown for SJ521054 and analogs (e.g. a pulldown experiment using 100 uM compound), but data for SJ1008066 appears missing.

We provide new orthogonal assay data in Fig. 5h that shows SJ1008066 as well as SJ521054 inhibit MAGE-A11 binding to PCF11 in cell lysates. Notably, we also show this is specific in that these compounds do not disrupt binding of MAGE-A3 or MAGE-F1 with their substrates in cell lysate co-IP experiments (Fig. 5i).

3. Additional direct and selective small molecule experiments, as noted above, are especially important since the 4-aminoquinoline series, including SJ1008066 are rather ‘feature-less’ and might be prone to promiscuous inhibition of hydrophobically-driven protein:protein and protein:peptide interactions.

We have performed Nano-differential scanning fluorimetry (NanoDSF) analysis to determine whether small compounds SJ521054 and SJ1008066 directly bind to MAGE-A11. Indeed, we find both compounds showed stabilization of MAGE-A11 in a six-point dose response confirming on-target activity of each compound. These data are now included in Fig. 5g and Extended data Fig. 5b-c.

Additionally, we show both SJ521054 and SJ1008066 compounds do not disrupt interaction of other MAGEs with their substrates (MAGE-A3:AMPKa1 and MAGE-F1: MMS19) in cell lysates (in Fig.5i). In combination, these results suggest that 1) the compounds directly bind MAGE-A11 MHD to disrupt binding to PCF11 and 2) this is not promiscuous inhibition, but rather specific to MAGE-A11.

4. Above confirmatory experiments are preferred and desired for publication of the small molecule section of the manuscript. But, it is unclear why small molecule hits weren’t at least crudely tested with differential scanning fluorimetry since that method was used for peptides as shown in Extended data Figure 2. Also, it would be desirable to show at least some representative examples of thermal melts in extended data.

As suggested we performed NanoDSF analysis in Fig. 5g and Extended data Fig. 5b-c. As described above, the results are consistent with our previous conclusions of direct binding between the small molecules and MAGE-A11 MHD.

5. Additional selectivity data for SJ1008066 amongst other MAGE family members would significantly add to the manuscript. Either result, that the small molecule inhibitor is selective or pan-MAGE, would be very interesting. It is understood that a validated small molecule inhibitor may potentially be MAGE-specific given the MAGE family discussion herein and data shown in Figure 4, however some supportive data would be an improvement.

We appreciate this thoughtful suggestion. We have now tested whether SJ521054 or SJ1008066 disrupt binding between MAGE-A3 with its substrate AMPKa1 or MAGE-F1 with MMS19 by co-IP experiments in cell lysates. Unlike MAGE-A11:PCF11 (Fig. 5h), we find that SJ521054 and SJ1008066 do not disrupt interaction of MAGE-A3 or MAGE-F1 with their substrates AMPKa1 or MMS19, respectively (Fig. 5i). These results suggest that both small compounds selectively target MAGE-A11 and not all MAGEs in general.

6. Were traditional peptide co-crystal structures attempted and failed leading to the MAGE-A11:PCF11 peptide fusion construct? Including more information here would be beneficial to readers.

Co-crystal structures of MAGE-A11 MHD with PCF11 peptide either fused to MAGE-A11 or added in solution post-protein purification were tested in parallel. We found that providing the PCF11 peptide in solution did not yield homogenous solution of MAGE-A11 MHD:PCF11 compared to the fused approach. This is likely due to the apo-protein being less stable during expression and purification from bacteria. Thus, having the native peptide present during expression and purification helps stabilize the protein. This information has now been included in the results section (p.7).

7. Concerning the proteome search using the consensus MAGE-A11 interacting peptide, it might be valuable to additionally search for where the peptide motif is located in the protein if a crystal structure exists. For example, the lack of confirmation that MTPN (myotrophin) interacts with MAGE-A11 is likely explained by the fact that the residues shown to be important for binding are not solvent exposed (brief analysis of PDB: 3AAA). In fact, it is often observed that linear sequence similarity to a short peptide motif does not correlate to presentation of that motif on a properly folded protein. This a rather minor point, but some additional discussion/analysis here would further support the approach taken and the observation that RACGAP1, ANKRD65 and ZBTB7C are bona fide MAGE-A11 interaction partners. Also, it is suggested that the word 'robust' be changed or removed (page 7, line 136) since only a crude pulldown experiment from IVT was completed. Definitive confirmation would require an alternate and more quantitative assay format and competition studies with PCF11 peptide.

We appreciate the insightful comments and analysis. Indeed, our sequence search did not take into account surface exposure of the binding motif. We have noted this limitation through highlighting that MTPN likely fails to bind due to buried motif based on the structure (PDB: 3AAA). Thank you for pointing this out. It provides additional evidence for our conclusions. Additionally, as suggested, 'robust' has been removed.

Very minor typos/edits:

1. Comma after Finally; page 16, line 349.

Corrected.

2. 200 n of ligand PCF11_6; page 24, line 537.

Corrected.

3. Missing period: acids Notably, the PCF11 sequence; line 131, page 7.

Corrected.

4. Either change to 'agonists' (actually, think this small molecule is an 'antagonist' for IL-2 binding or add 'a' and 'inhibits': including small molecule agonist that inhibit T-cell proliferation; lines 251/252, page 12.

Corrected as suggested.

5. Double check screening metrics throughout text and figures: for example, ED Fig. 4a states '15 4-aminquinolines studied' and ED Table 5 title (Extended Data Table 5: TR-FRET and ADME results of 14 4-Aminoquinoline compounds) shows 14 compounds (text states 9 additional compounds).

Corrected. Extended Data Table 5 title changed to: TR-FRET, solubility and permeability studies of 4-Aminoquinoline compounds. The specific numbers stated are now correct. Any discrepancies are not mistakes, but rather due to overlap in compounds between different screening decks.

6. Seems that 'proteins' is more appropriate here: 'the underlying function and biochemical activity of MAGE genes'; page 14, line 296

Changed as suggested.

7. Choice of the word 'substrate' for the binding cleft is a little unusual since MAGEs are not enzymes.

We agree that MAGEs are not enzymes. However, it is common nomenclature within the ubiquitin field that E3 ubiquitin ligase targets are referred to as 'substrates' and thus the modular components of the ligases that bind the substrate are often referred to as the 'substrate binding' or 'substrate selectivity factors' with the pockets/clefts being referred to as the 'substrate binding pocket/cleft'. Thus, we believe that our description is consistent with the field's nomenclature and definitions.

8. ADME in section title and table title really is not appropriate for solubility and Caco-2 permeability studies.

Changed the title of Extended Data Table 5 as suggested to: TR-FRET, solubility and permeability studies of 4-Aminoquinoline compounds.

Reviewer #3: MAGE proteins serve as substrate adapters for single polypeptide RING and HECT E3 ligases thereby augmenting their substrate scope. MAGE proteins can drive cancer and this is exemplified by the protein MAGE-A11. How MAGE proteins recognise substrates is unknown and therapeutic disruption of MAGE-substrate interactions could have clinical value. The manuscript by Yang et al. identifies a substrate binding cleft within the E3 ligase adapter protein MAGE-A11. A minimal degron within the MAGE-A11 substrate PCF11 is subsequently mapped using a peptide library coupled to a TR-FRET binding assay. A general recognition motif is determined and searched against the human proteome. Disruption of the MAGE binding cleft is also shown to ablate substrate recognition in cells. A crystal structure is solved revealing the structural basis for recognition. Finally, a HTS is carried out and proof of concept inhibitors are identified. The assays, structural work and HTS are carried out to a high standard and on the whole the data are well interpreted.

We appreciate the effort and time that reviewer provided to improve our study and to provide insightful feedback and comments on our manuscript. Below we address the critiques raised, including additional experimental evidence.

I have the following major concerns:

1. “Given the importance of the C-terminal end of the PCF11 motif for MAGE-A11 binding and that MAGE-A11:PCF11 interaction affinity is not fully optimized, we synthesized several peptides with non-natural phenylalanine derivatives at PCF11 F693.” It is not entirely clear why the C-terminal end of the peptide was targeted for non-canonical amino acid mutagenesis.

We observed that 5 positions within the 12 amino acid PCF11 motif are critical for binding to MAGE-A11. Strikingly, 4 of these residues are on the C-terminal half of the 13 amino acid sequence (positions #8, #9, #11, and #12) (Fig. 1g,h). Thus the C-terminal end of the peptide seems to contribute the most to binding. This is why the C-terminal end of the peptide was targeted. Additional rationale as discussed above has been added to p.9 to provide clarity.

What is meant by the interaction affinity not being fully optimized?

Our apologizes for the confusion. We observed that the affinity of the natural PCF11 motif can be improved upon by amino acid substitution (natural or unnatural) at specific positions (Fig. 1h). We have reworded the sentence on p.9 to remove the word optimized and clear up confusion.

2. SJ521054 binding to MAGE-A11 should be quantified with a complementary biophysical technique such as ITC or SPR.

We understand the reviewer’s concern and agree that biophysical experiments would provide additional support for direct binding of small molecules to MAGE-A11. We have tried preferable approaches, such as ITC or SPR to address these issues. We faced challenges with expression of enough high quality apo-MAGE-A11 protein that precluded

us from conducting ITC experiments. For SPR, His-MAGE-A11 protein appeared to load onto the chips at a sufficient level, but injection of ligands, including positive control PCF11 peptides, led to erratic sensorgrams, indicative of poor stability on the chip or interactions with the chip surface. As an alternative, we were able to perform NanoDSF to provide evidence of direct binding of both SJ521054 and SJ1008066 to MAGE-A11 MHD (Fig. 5g and Extended Data Fig. 5b,c). We believe that these data provide orthogonal validation that the small molecules directly bind MAGE-A11.

3. Please provide a representative blot for the quantification In figure 5d to make this more convincing.

Our apologies, this is now shown in Fig. 5h. We have also added additional data showing both SJ521054 and SJ1008066 disrupt MAGE-A11:PCF11 interaction. Additionally, this is specific as neither compound can disrupt interaction of MAGE-A3 or MAGE-F1 with their substrates (Fig. 5i).

4. Can the authors explain/discuss why when in 5d 100 uM compound in cell lysates reduces MAGE-A11-PCF11 binding 2-fold yet in 5f, 20 uM is sufficient to reduce viability to near zero?

These assays are very different. One measures binding of the two proteins in a cell lysate, whereas the other monitors cell viability. We would not expect these two assays to necessarily be perfectly aligned on the amount of compound needed to observe an effect. One can think of many reasons why this may be: 1) Decreasing the binding of MAGE-A11 by only a fraction (say around 50%) may indeed be sufficient to cause cytotoxicity. This would be consistent with our previous results in which siRNAs that only partially deplete MAGE-A11 show decreased cell viability; 2) the measurements happen in very different contexts, one in live cells and the other in concentrated cell lysates. Thus, we would indeed expect that the high concentration of cell lysates would show reduced capacity for the small molecules to disrupt MAGE-A11:PCF11 interaction due to the large number of other potential off-target interactions that sequester the compounds; and 3) the compounds just may not be as active in vitro. This could be caused by a number of reasons, including being metabolized in cells. Regardless of the exact reasons, we believe that the results presented provide a compelling case for our conclusions.

Also in a MAGE-A11 independent manner?

Our data support that at lower concentrations, SJ521054 induced decreased cell viability in a MAGE-A11-dependent manner. We show this in both DAOY and HEK293 cells (Fig. 4e and Extended Data Fig. 4f). Notably, this is not dependent on changes in proliferation rate of the cells with or without MAGE-A11, as shown in Extended Data Fig. 4g. However, it is true that at higher concentrations SJ521054 shows cytotoxicity to both cell lines in a MAGE-independent manner.

5. The AlamarBlue assay used in 5f measures proliferation, similarly to the data presented in 3b. In 3b KO cell proliferation is reduced 2-fold relative to A11 overexpressing cells. Therefore in 5f, it would be expected that the proliferation of the vector control cells is also reduced ~2-fold

relative to A11 overexpressing lines. Therefore, It is not clear if the A11 dependence is wholly due to compound binding to A11 or increased assay signal as a result of the higher proliferation rate.

Correcting for proliferation differences isn't feasible in DAOY KO cells. Thus, we turned to HEK293FT cells. We monitored cytotoxicity and proliferation in HEK293FT cells with or without MAGE-A11 expression. We noted no changes in cell proliferation upon expression of MAGE-A11 (Extended Data Fig. 4g). However, MAGE-A11 expressing HEK293FT cells were more sensitive to SJ521054 (Extended Data Fig. 4f). These results are consistent with the conclusion that the increased sensitivity of MAGE-A11-positive cells to SJ521054 is due to on-target effect of inhibiting MAGE-A11 and not due to differences in cell proliferation rate.

6. Is MAGE-A11 monomeric in solution? This could be readily tested by SEC-MALS for example.

We observed no evidence of dimerization in our crystal structure or by SEC, suggesting that MAGE-A11 is not a dimer. To further test this in cell lysates, we used two differentially tagged MAGE-A11 proteins and performed co-IP. We found that Flag-MAGE-A11 did not co-IP with Myc-MAGE-A11, suggesting that MAGE-A11 does not form dimers in cell lysates (Extended Data Figure 2a). Although we did not have access to SEC-MALS, we believe these data in combination are consistent with the conclusions.

Minor concerns:

1. There are numerous typos throughout e.g phenylalanine is frequently incorrect.
Changed to phenylalanine.

The supplemental also has numerous typos e.g: 2. Page 2 Line 5: chromatograpy
Changed to chromatography.

3. Page 3 line 8, Page 5 line 22: hexanes-ethyl actate
Changed to hexanes-ethyl acetate.

4. Page 4 line 14: sodium borodydride
Changed to sodium borohydride.

5. Page 9 line 27 and throughout: Biotage 11g NH colomn
Changed to Biotage 11g NH column.

REVIEWERS' COMMENTS:

Reviewer #1 (Remarks to the Author):

In my opinion this is a major step forward in MAGE biology. The Authors have identified a novel mechanism that targets MAGE-11 oncogenic activity and they have shown it is specific for MAGE-11 and represents a potentially drugable target. The mechanism appears conserved in MAGE in general but is specific for individual substrate-ligand pairs. They have tested their hypotheses in vitro and in vivo and confirmed the structures involved. They have answered the questions from the previous review.

Reviewer #2 (Remarks to the Author):

The authors have responded adequately to reviewers comments.

Reviewer #3 (Remarks to the Author):

The reviewers have satisfactorily addressed all concerns and I recommend publication.

REVIEWERS' COMMENTS:

We thank the reviewers for their time peer reviewing our work and providing insightful suggestions along the way.

Reviewer #1 (Remarks to the Author):

In my opinion this is a major step forward in MAGE biology. The Authors have identified a novel mechanism that targets MAGE-11 oncogenic activity and they have shown it is specific for MAGE-11 and represents a potentially drugable target. The mechanism appears conserved in MAGE in general but is specific for individual substrate-ligand pairs. They have tested their hypotheses in vitro and in vivo and confirmed the structures involved. They have answered the questions from the previous review.

Reviewer #2 (Remarks to the Author):

The authors have responded adequately to reviewers comments.

Reviewer #3 (Remarks to the Author):

The reviewers have satisfactorily addressed all concerns and I recommend publication.